# Phase separation and zinc-induced transition modulate synaptic distribution and association of autism-linked CTTNBP2 and SHANK3

Pu-Yun Shih[1,4,6], Yu-Lun Fang[1,2,6], Sahana Shankar[1,3], Sue-Ping Lee[1], Hsiao-Tang Hu[1], Hsin Chen[1,5], Ting-Fang Wang [1,3], Kuo-Chiang Hsia [1,3✉] & Yi-Ping Hsueh [1,3✉]

Many synaptic proteins form biological condensates via liquid-liquid phase separation (LLPS). Synaptopathy, a key feature of autism spectrum disorders (ASD), is likely relevant to the impaired phase separation and/or transition of ASD-linked synaptic proteins. Here, we report that LLPS and zinc-induced liquid-to-gel phase transition regulate the synaptic distribution and protein-protein interaction of cortactin-binding protein 2 (CTTNBP2), an ASD-linked protein. CTTNBP2 forms self-assembled condensates through its C-terminal intrinsically disordered region and facilitates SHANK3 co-condensation at dendritic spines. Zinc binds the N-terminal coiled-coil region of CTTNBP2, promoting higher-order assemblies. Consequently, it leads to reduce CTTNBP2 mobility and enhance the stability and synaptic retention of CTTNBP2 condensates. Moreover, ASD-linked mutations alter condensate formation and synaptic retention of CTTNBP2 and impair mouse social behaviors, which are all ameliorated by zinc supplementation. Our study suggests the relevance of condensate formation and zinc-induced phase transition to the synaptic distribution and function of ASD-linked proteins.

[1] Institute of Molecular Biology, Academia Sinica, Taipei, Taiwan, ROC. [2] Department and Graduate Institute of Biochemistry, National Defense Medical Center, Taipei, Taiwan, ROC. [3] Molecular and Cell Biology, Taiwan International Graduate Program, Institute of Molecular Biology, Academia Sinica and Graduate Institute of Life Sciences, National Defense Medical Center, Taipei, Taiwan, ROC. [4] Present address: Department of Neurology, University of California San Francisco, San Francisco, USA. [5] Present address: Undergraduate Program in Neuroscience, John Hopkins University, Baltimore, USA. [6] These authors contributed equally: Pu-Yun Shih, Yu-Lun Fang. ✉email: khsia@gate.sinica.edu.tw; yph@gate.sinica.edu.tw

Liquid–liquid phase separation (LLPS) is a physical process by which proteins and nucleic acids are condensed into liquid droplets (also termed membraneless or biological condensates) via interactions among multivalent binding proteins or intrinsically disordered proteins (IDP)/nucleic acids[1,2]. LLPS is involved in forming membraneless organelles, enzymatic assemblies and functionally relevant protein macrocomplexes, all of which play roles in diverse cellular processes such as transcription, splicing, translation, and transportation[3–7]. In neurons and muscles, LLPS is relevant to Tau- and TDP43-related neurodegenerative diseases and RAPSN-related congenital myasthenic syndrome[8–11]. Given that LLPS also influences protein assembly at the presynaptic active zone and postsynaptic density to regulate neurotransmission[5,11–17], it is also likely involved in synaptopathies.

Autism spectrum disorders (ASD), a group of neurodevelopmental disorders, are recognized as a synaptopathic condition[18–23] involving abnormal neural circuits[24–27]. Protein products of several ASD-linked genes, including SYNGAP1 and SHANK3, have been shown to interact with other postsynaptic proteins such as PSD-95 and stargazin to form liquid droplets in vitro[12,28]. Synapsin 1 and other synaptic proteins in presynaptic terminals also accumulate at the active zone via LLPS[14–16,29]. However, the physiological relevance of LLPS for ASD-related synaptopathies has remained elusive.

*Cortactin-binding protein 2* (*CTTNBP2*), an ASD-associated gene[20–22,30], encodes a neuron-specific cytoskeleton-associated protein[31–34] that controls dendritic spine formation and maintenance[32,33,35,36]. Secondary structure analysis on amino acid (aa) sequences indicates that CTTNBP2 consists of an N-terminal coiled-coil domain and a C-terminal intrinsically disordered region (IDR)[36] (Fig. 1a). The coiled-coil domain mediates CTTNBP2 oligomerization, including homo-oligomerization and hetero-oligomerization with striatins[33], a regulatory subunit of the protein phosphatase 2A family. The C-terminal IDR comprises two elements; a middle domain responsible for microtubule interactions, and a proline-rich region of which the last ~100 aa interacts with cortactin[31,32,36].

In immature neurons, CTTNBP2 forms puncta in the soma and along dendrites. It interacts with and bundles microtubule cytoskeletons to promote dendritic arborization[34]. In mature neurons, CTTNBP2 is highly concentrated at dendritic spines, where it interacts with cortactin-F-actin cytoskeletons to control cortactin mobility[32,33]. High potassium or glutamate stimulation triggers relocalization of F-actin, cortactin and striatins, but not CTTNBP2, from dendritic spines to the dendritic shaft[32,33], suggesting a potential role for CTTNBP2 in marking dendritic spines during synaptic remodeling in response to neurotransmission. Recently, a proteomic analysis showed that *Cttnbp2* knockout alters the synaptic distribution of more than 100 proteins[35]. Some CTTNBP2-regulated proteins, such as SHANK3, SHANK2, STRN (striatin) and RAC3, physically associate with CTTNBP2[35]. Thus, CTTNBP2 acts as a multi-domain adaptor molecule to control F-actin and microtubule cytoskeletons and multiple synaptic molecules[37]. However, it remains unclear how CTTNBP2 interacts with and controls the distribution of many different synaptic proteins.

Human genetic studies have identified several ASD-linked mutations in the *CTTNBP2* gene[20–22,30]. Among them, we have previously demonstrated that the M120I, R533* (where * represents a stop codon) and D570Y mutations of the murine *Cttnbp2* gene differentially impact the protein–protein interactions of CTTNBP2[36]. Both the M120I and R533* mutations reduced or completely disrupted interactions between CTTNBP2 and cortactin, whereas the D570Y mutation enhanced microtubule binding by CTTNBP2[36]. Note that residue M120 is not located in the cortactin binding site of CTTNBP2 and residue D570 is not

sited in the microtubule-binding region[34,36]. Thus, these results imply that complex and flexible intramolecular interactions regulate interaction of CTTNBP2 with its binding partners. Although these ASD-linked mutations exert differential impacts on CTTNBP2, they all result in a similar synaptopathic phenotype, namely a reduced density and size of dendritic spines[36], both key features of ASD. We have also previously generated and characterized various *Cttnbp2*-deficient/mutated mouse lines. These *Cttnbp2*[+/−], *Cttnbp2*[−/−], *Cttnbp2*[+/R533*] and *Cttnbp2*[+/M120I] mice all exhibit synaptopathies[35,36]. Moreover, these mice also display autism-like behaviors, such as social deficits in three-chamber and reciprocal social interaction tests, abnormal ultrasonic vocalization, cognitive inflexibility in a T-maze, and memory impairment in a novel place recognition test[35]. Thus, deletion or ASD-linked mutations of *Cttnbp2* indeed impair synapse formation/maintenance and result in behavioral abnormalities.

Our previous studies have also suggested the relevance of zinc to CTTNBP2 function. First, *Cttnbp2* knockout reduced the synaptic distribution of 18 zinc-binding or -regulated proteins[35]. Moreovoer, zinc levels are reduced in *Cttnbp2* knockout mouse brains[35]. Conversely, CTTNBP2 overexpression in COS1 cells lacking endogenous CTTNBP2 enhanced intracellular zinc concentrations[35]. Furthermore, zinc supplementation enhanced the synaptic distribution of CTTNBP2-regulated proteins and ameliorated the behavioral deficits of *Cttnbp2*[−/−] and *Cttnbp2*[+/R533*] mice[35]. Thus, zinc is highly relevant to the functions of CTTNBP2 in neurons and brains, though it is unclear how zinc and CTTNBP2 influence each other.

CTTNBP2 immunoreactivities always manifest as a punctate pattern, no matter whether assessed at the dendritic spines of mature neurons or along microtubules in immature neurons[32,34]. Given that almost two-thirds of the CTTNBP2 protein is intrinsically disordered, we speculate that CTTNBP2 aggregates in neurons are biological condensates. In this study, we demonstrate that CTTNBP2 forms mobile condensates via its C-terminal region and that zinc binding to the N-terminal region induces a liquid-to-gel phase transition. We further reveal the impact of LLPS and zinc on the synaptic distribution of CTTNBP2 at dendritic spines using cultured hippocampal neurons. Zinc retains CTTNBP2 at dendritic spines. SHANK3, another zinc-binding synaptic protein, co-localizes with CTTNBP2 condensates, via processes involving LLPS and zinc. Moreover, ASD-linked CTTNBP2 mutations alter LLPS and the synaptic distribution of CTTNBP2. However, all ASD-linked CTTNBP2 mutant variants still responded to zinc, providing a potential explanation for how zinc supplementation ameliorates the neuronal and brain defects caused by *Cttnbp2* deficiency and is consistent with a role for zinc deficiency in ASD etiology. In conclusion, our study supports that LLPS represents the mechanism by which CTTNBP2 condenses in dendritic spines and that zinc serves as a signal to immobilize and retain CTTNBP2 there by inducing the liquid-to-gel phase transition of CTTNBP2. Overall, we provide evidence for how LLPS and liquid-to-gel phase transitions control synaptic retention and the protein–protein interactions of ASD-linked proteins.

## Results

**CTTNBP2 forms membraneless condensates.** We applied Jpred4 (https://www.compbio.dundee.ac.uk/jpred/)[38] and Robetta (https://robetta.bakerlab.org) to predict structural features of the short form of CTTNBP2 (i.e., the first 630 amino acids (aa)), since it is the sole form expressed in the brain[32,33,35]. They both predicted residues 104–271 as a long α-helix, as well as revealing several short α-helixes within the first 100 aa or in

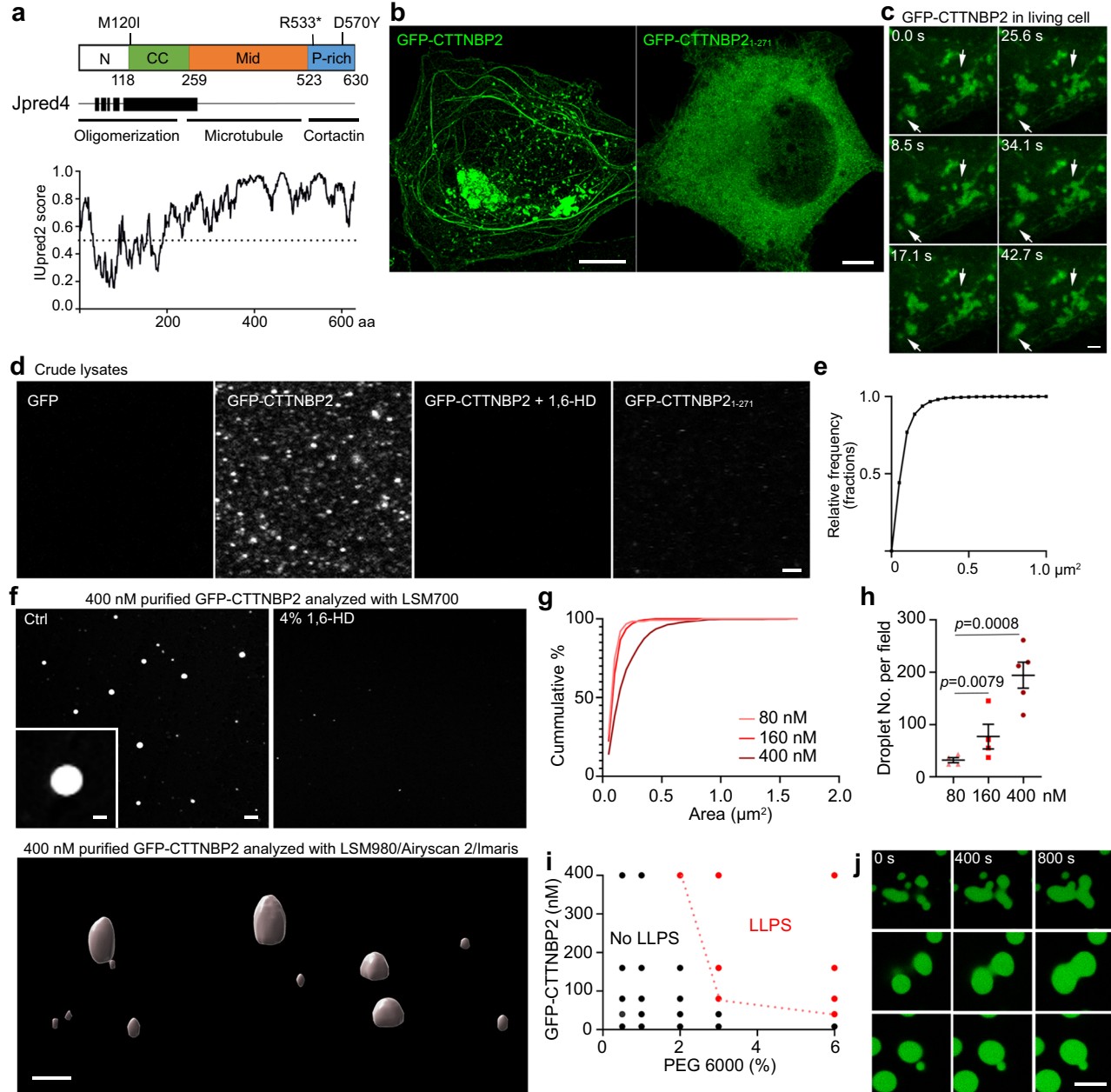

**Fig. 1 CTTNBP2, an intrinsically disordered protein, undergoes LLPS in cellulo and in solution. a** Predictions of CTTNBP2 protein structure. Upper: Schematic of CTTNBP2 protein domains. Three selected ASD-linked mutations, i.e. M120I, R533* and D570Y, are highlighted. Middle: Jpred4-predicted α-helical secondary structures are shown as black blocks. CTTNBP2 domains and corresponding functions are indicated. Bottom: IUpred2A-predicted intrinsic disorder of CTTNBP2. Higher scores indicate a greater degree of disorder. **b** Confocal images of COS1 cells expressing GFP-CTTNBP2 and GFP-CTTNBP2$_{1-271}$. **c** Time-lapse imaging of GFP-CTTNBP2 in a COS1 cell. Arrows point to protein condensates undergoing fusion and fission. **d** Droplet-like condensate formation assay of crude lysates from GFP-CTTNBP2-expressing HEK293T cells with 1.5% PEG in the presence or absence of 8% 1,6-HD. Representative confocal images are shown. **e** Cumulative probability of condensate size of GFP-CTTNBP2 in HEK293T crude lysate. **f** Condensate formation by purified GFP-CTTNBP2 proteins. Upper: Confocal images of CTTNBP2 condensates captured by LSM700 microscopy. Lower: 3D modeling of CTTNBP2 condensates acquired by an LSM980 system with Airyscan 2 and analyzed in Imaris software. GFP-CTTNBP2 concentration = 400 nM. **g** Dosage effect of purified GFP-CTTNBP2 protein on condensate size. $N = 971$ (400 nM), 308 (160 nM) and 128 (80 nM) condensates over two independent reactions. **h** Dosage effect of purified GFP-CTTNBP2 protein on condensate number. Data were collected from $N = 5$ (400 nM), 4 (160 nM), 4 (80 nM) images. **i** Phase diagram of purified GFP-CTTNBP2 confirmed by two independent reactions. **j** Fusion of purified GFP-CTTNBP2 condensates. Purified GFP-CTTNBP2 (10 µM) with 3% PEG was mixed and observed in real-time using an LSM980 system with Airyscan 2. Three examples are shown at 400-s intervals. In **h**, one-way ANOVA with Bonferroni's multiple comparisons test. All statistical analysis and results are summarized in the Statistic Information in the source data file. Scale bars: **b**, **j** 5 µm; **c** and **d** 2 µm; **f** upper, 2 µm; inset, 0.4 µm; lower, 1 µm.

the C-terminal tail (Fig. 1a, S1), echoing previous findings that the N-terminal region of CTTNBP2 contains an oligomerizing coiled-coil domain[33,34]. The remaining C-terminal region of CTTNBP2 mainly comprises a random coil, which exists in an open conformation or folds back to interact with the N-terminal coiled-coil region (Fig. S1), corroborating our previous finding that the N-terminal coiled-coil domain co-immunoprecipitates with the C-terminal domain of CTTNBP2[36].

IUpred2 (https://iupred2a.elte.hu) was then used to predict that the C-terminal region (200–630 aa) of CTTNBP2 is an intrinsically disordered region (IDR), with an IUpred score consistently >0.5 (Fig. 1a).

As IDRs are frequently involved in forming condensed biological assemblies, i.e. membraneless condensates, we investigated if CTTNBP2 undergoes LLPS to form such condensates. First, we used transfected COS1 and HEK293T cells that do not express endogenous CTTNBP2 to investigate the properties of CTTNBP2 proteins in cells. Consistent with our previous findings[33,34], GFP-CTTNBP2 formed irregular condensates or filaments along microtubules in transfected COS1 cells (Fig. 1b) and HEK293T cells (Fig. S2a). Next, we used GFP-CTTNBP2$_{1-271}$, which possesses the entire N-terminal α-helix but lacks the C-terminal IDR, to demonstrate that the C-terminal IDR of CTTNBP2 is required to form condensates because GFP-CTTNBP2$_{1-271}$ did not form obvious condensates in COS1 cells, instead being quite evenly distributed (Fig. 1b).

We further perfomed live-imaging of both COS1 and HEK293T cells to show that the GFP-CTTNBP2 condensates were dynamic, exhibiting constantly changing morphology, with some condensates rapidly undergoing fusion and fission (Figs. 1c, S2b, S2c; Movies S1 and S2). When we photobleached almost all the GFP signal of a single condensate, within a few minutes the GFP signal in the condensate rapidly increased and was distributed throughout the entire condensate (Fig. S2d). During recording, this particular condensate did not fuse with other condensates, suggesting rapid molecular exchange between the condensate and the neighboring cytoplasm. These results support that CTTNBP2 forms dynamic and membraneless condensates in cells.

Next, we used crude lysates of transfected HEK293T cells to investigate the condensate formation ability of CTTNBP2 in solution. In the presence of polyethylene glycol (PEG), a commonly used crowding agent[39], GFP-CTTNBP2 readily formed droplets as large as $0.1\,\mu m^2$ or greater (Figs. 1d, 1e). These PEG-induced droplets were specifically attributable to CTTNBP2 because GFP alone did not form droplets in PEG solution (Fig. 1d). GFP-CTTNBP2 condensates were sensitive to treatment with 1,6-hexanediol (1,6-HD), an aliphatic alcohol that disrupts weak hydrophobic protein–protein interactions and dissolves protein droplets formed by IDP[8,9,40] (Fig. 1d). Moreover, GFP-CTTNBP2$_{1-271}$ did not form condensates in solution (Fig. 1d). These results suggest that GFP-CTTNBP2 forms condensates in cells via its C-terminal IDR.

We further subjected purified GFP-CTTNBP2 proteins (Fig. S3a, S3b) to a condensate formation assay. At a concentration of $0.4\,\mu M$ or less, purified GFP-CTTNBP2 proteins also formed 1,6-HD-sensitive spherical droplets in the presence of PEG (Fig. 1f). The number and size of CTTNBP2 condensates were also correlated with the concentration of GFP-CTTNBP2 (Fig. 1g–i). Time-lapse recording further revealed that the droplets formed by purified GFP-CTTNBP2 readily fuse to form bigger condensates in solution (Fig. 1j, Movies S3–S5). Together, these results indicate that purified CTTNBP2 proteins form biological condensates via LLPS in solution.

Thus, these results indicate that CTTNBP2 undergoes LLPS to form condensates, which are sensitive to 1,6-HD treatment.

**CTTNBP2 is a zinc-binding protein.** The N-terminal α-helix (1–271 aa) contains 56 Glu or Asp and 9 His or Cys residues, with a predicted isoelectric point of ~5.0–5.4. Since a negatively charged α-helix may chelate zinc[41,42], we speculated that CTTNBP2 can bind zinc via its negatively charged N-terminal coiled-coil domain. To test that possibility, we purified CTTNBP2 fragments comprising residues 1–271 and 104–271, respectively,

denoted CTTNBP2$_{1-271}$ and CTTNBP2$_{104-271}$, and subjected the CTTNBP2 fragments to three different experiments to investigate their interactions with zinc.

First, we employed circular dichroism (CD) spectrometry to assess the absorption wavelength of CTTNBP2$_{104-271}$ in the presence of 0, 0.15, or 0.3 mM zinc. CD represents the differential absorption of circularly polarized light by optically active matter in solution[43,44]. Different types of secondary structures exhibit characteristic features in CD spectra. CTTNBP2$_{104-271}$ presented a major negative peak at ~225 nm, implying that it forms an α-helix (Fig. 2a). Elevating the zinc concentration to 0.15 or 0.3 mM induced a shift from ~225 to ~235 nm, indicative of changes in the protein secondary structure (Fig. 2a). Additionally, as the depth of the curve (quantified as $deg\,cm^2\,dmol^{-1}$) in CD spectra is directly proportional to the concentration of protein in solution, the reduced peak height indicates a reduced concentration of CTTNBP2, most likely due to protein aggregation. To confirm this speculation, we compared CTTNBP2$_{104-271}$ protein amounts in the precipitates and supernatants, which were separated by centrifugation. Higher zinc concentration indeed resulted in more CTTNBP2$_{104-271}$ precipitates (Fig. 2b), supporting that zinc binds and induces higher-order multimerization of CTTNBP2 oligomers to exacerbate protein precipitation. Next, we compared the effect of 1 mM of zinc ions to that of other ions at the same concentration to investigate if the conformational change in CTTNBP2$_{104-271}$ is zinc-specific. We found that other metal ions, including copper, iron, and potassium, did not induce a shift in wavelength, though peak height was reduced by copper and iron in the CD spectra (Fig. 2c), suggesting that 1 mM copper or iron can induce protein precipitation of CTTNBP2 but not a conformational change.

Second, we applied a zinc-chelating resin to purify zinc-binding proteins. Compared to resin lacking zinc, the precharged zinc resin bound more CTTNBP2$_{1-271}$ from solution (Fig. 2d). A previous study demonstrated that the SHANK3 SAM domain binds zinc[45], and our precharged zinc resin also bound the SHANK3 SAM domain (Fig. 2d), supporting the reliability of our analysis.

Third, isothermal titration calorimetry further revealed a binding affinity ($K_d$) of CTTNBP2$_{104-271}$ for zinc of $7.31 \pm 1.05\,\mu M$ (Fig. 2e), suggesting a micromolar binding affinity between CTTNBP2 and zinc. Notably, the Kd$_{Zn}$ of CTTNBP2$_{104-271}$ is lower than the concentration of zinc in serum (~15 μM), total brain (~150 μM) and synaptic vesicles (~1 mM)[46], allowing CTTNBP2 to bind zinc under physiological conditions.

**Zinc induces the liquid-to-gel phase transition of CTTNBP2.** The above-described experiments indicate that zinc binds and induces high-order multimerization or even precipitation of CTTNBP2. Thus, we postulated that zinc influences the protein phase of CTTNBP2. To explore that possibility, we added zinc at a final concentration of 0.1 mM to HEK293T cell lysate containing GFP-CTTNBP2. Zinc dramatically induced big aggregations ($>1 \times 1\,\mu m$ in the $X-Y$ dimension) of CTTNBP2, even in the presence of 1,6-HD (Figs. 2f, S1c). This effect was zinc-specific because copper did not have the same impact (Fig. 2f). Addition of zinc to purified GFP-CTTNBP2 also induced large irregular aggregates in the presence of PEG (Fig. 2g, h, Movies S6 and S7). Thus, whether considering free oligomers or condensates, zinc clearly induces CTTNBP2 aggregation.

We further investigated if zinc-induced CTTNBP2 aggregation is reversible. Thirty min after incubating purified GFP-CTTNBP2 proteins with zinc, we added TPEN (N,N,N′,N′-Tetrakis(2-pyridylmethyl)ethylenediamine, a zinc chelator) and incubated for a further 30 min. As the Kd$_{Zn}$ of TPEN is at the fM level[47] and

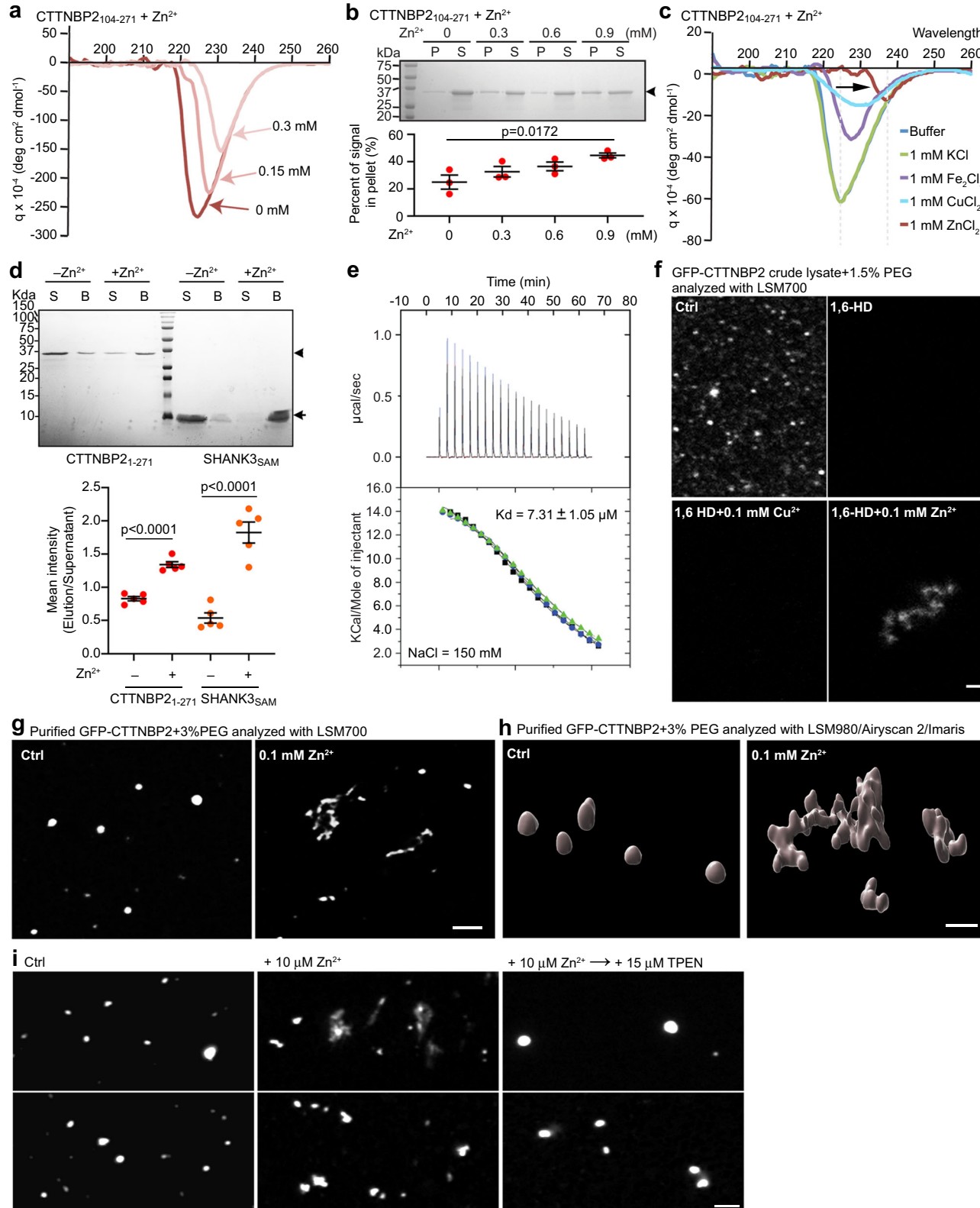

the $Kd_{Zn}$ of CTTNBP2 is at μM level (Fig. 2e), TPEN would be expected to effectively compete for zinc binding with CTTNBP2. Indeed, the irregular CTTNBP2 aggregates disappeared upon TPEN treatment (Fig. 2i). Thus, CTTNBP2 condensates dynamically change in response to zinc concentration.

Overall then, zinc binds to CTTNBP2 and reversibly induces aggregation of CTTNBP2 oligomer and condensates.

**ASD-linked mutations and zinc regulate CTTNBP2 dynamics.** Our previous studies have shown that certain ASD-linked mutations alter the functions of CTTNBP2[35,36]. If phase separation and zinc binding are indeed critical properties of CTTNBP2, we expected that certain ASD-linked mutations may alter those properties. To investigate that possibility, first we used IUpred2 to assess various ASD-linked CTTNBP2 mutants

**Fig. 2 Zinc directly binds and regulates CTTNBP2 phase transition. a** Circular dichroism (CD) spectra of CTTNBP2$_{104-271}$ in the absence or presence of zinc at the indicated concentrations. **b** SDS–PAGE analysis of CTTNBP$_{1-271}$ sedimentation in the absence or presence of zinc at the indicated concentrations. Supernatant and pellet fractions are indicated as S and P, respectively. CTTNBP2$_{1-271}$ band intensities in the gel were quantified and plotted. Standard deviation was determined from three independent experiments. Two-tailed Student's $t$ test. **c** CD spectra of CTTNBP2$_{104-271}$ in the presence of the indicated metal ions (concentration of 1 mM). The zinc spectrum is indicated by an arrow. **d** Pulldown assays of CTTNBP2$_{1-271}$ and SHANK$_{sam}$. CTTNBP2$_{1-271}$ and SHANK$_{sam}$ were incubated with precharged zinc resin. Supernatant (S) and bound (B) samples were analyzed by SDS–PAGE and stained with Coomassie blue. CTTNBP2$_{104-271}$ and SHANK$_{sam}$ band intensities in the gel were quantified and plotted. Standard deviation was determined from three independent experiments. Two-tailed Student's $t$ test. **e** ITC titration curves (upper) and binding isotherms (lower) of CTTNBP2$_{104-271}$ with zinc. The $K_d$ value was determined from three independent experiments, as indicated. **f** Representative fluorescence image of HEK293T crude lysates expressing GFP-CTTNBP2 in the presence of 1,6-HD with or without Cu$^{2+}$ or Zn$^{2+}$, as indicated. **g** Representative fluorescence image of purified GFP-CTTNBP2 in the absence or presence of Zn$^{2+}$. **h** Imaris-generated 3D images of purified GFP-CTTNBP2 with or without Zn$^{2+}$. **i** Representative fluorescence image of purified GFP-CTTNBP2 in the presence of Zn$^{2+}$ with or without TPEN, as indicated. TPEN was added 30 min after zinc addition. All statistical analysis and results are summarized in the Statistic Information in the source data file. Scale bars: **f–h** 1 μm; **i** 2 μm.

(Fig. S4a). Among six ASD-linked mutants we assessed [A112T, M120I, G342R, P353A, R533* and D570Y], we focused on D570Y, R533* and M120I for the following reasons: (1) D570Y resulted in the greatest reduction of IUpred score; (2) R533* results in loss of the C-terminal tail, so it also likely influences CTTNBP2 structure; and (3) M120I mutation reduces N–C termini interaction[36], so we speculated it may indirectly control the degree of disorder in the C-terminal IDR of CTTNBP2 (Fig. S4a).

To assess these three CTTNBP2 mutant proteins, we employed three sets of experiments. The first experiment was fluorescence recovery after photobleaching (FRAP) to determine the mobility of CTTNBP2 condensates in COS1 cells (Fig. S4b, S4c). Wild-type (WT) CTTNBP2 condensates were relatively mobile because fluorescence signals of the GFP-tagged condensates were effectively recovered after photobleaching (Fig. 3a, S5a; Movie S8). The immobile fraction of GFP-WT was ~50% (Fig. 3a, S5a), but 30 min after zinc addition the immobile fraction increased to >80% (Fig. 3a, S5a; Movie S9). Consistent with the specificity of zinc binding to CTTNBP2, copper did not alter the mobility of GFP-WT condensates (Fig. 3a, S5a; Movie S10). These FRAP analyses demonstrate that GFP-WT forms mobile condensates and that zinc treatment reduces their mobility in cellulo.

We then used FRAP to compare our selected CTTNBP2 mutant proteins to WT. We first examined the GFP-D570Y mutant (Fig. S5a; Movies S11–S13). The recovery rate of the GFP-D570Y mutant was slower than that of WT, but its immobile fraction was similar to WT (i.e. ~50%) (Fig. 3b). Moreover, as for WT, zinc addition immobilized GFP-D570Y condensates (immobile fraction >80%) (Fig. 3b). Together, these results indicate that the D570Y mutation reduces CTTNBP2 mobility but does not alter its overall immobile fraction or its response to zinc.

FRAP did not reveal any change in CTTNBP2 mobility for the R533* mutant relative to WT (Fig. S5b). The GFP-M120I mutant displayed much greater mobility than WT proteins, so we had to apply shorter imaging intervals for recording (Figs. 3c, S5a; Movies S14–S16). Approximately 30% of GFP-M120I mutant proteins were immobile (Fig. 3c, right), i.e., much lower than for WT proteins, and zinc addition increased the immobile fraction to ~70% (Fig. 3c, right).

Taken together, our FRAP analysis indicates that condensates formed by ASD-linked CTTNBP2 mutants exhibit differential mobility and that zinc slows the mobility of both WT and mutant CTTNBP2.

**ASD-linked mutations regulate phase separation of CTTNBP2.** The second set of experiments was to dissect the impact of the selected ASD-linked mutations on condensate formation in solution. For these experiments using crude lysates of HEK293T cells, only one mutant was compared to WT for each set of experiments to ensure identical experimental conditions for

comparison. Three parameters were assessed: (1) cumulative probability of droplet area <1 μm² (representing the majority of droplets, >90%); (2) number of condensates >1 μm²; and (3) average area of all condensates.

In crude lysates, the average area of GFP-D570Y condensates was ~40% greater than WT condensates (0.39 vs. 0.27 μm²), yet the cumulative probability of condensate area <1 μm² and the number of condensates >1 μm² were comparable to GFP-WT (Fig. 4a–d). Both GFP-R533* and -M120I formed smaller droplet condensates than WT (Fig. 4a), as reflected by a lower cumulative probability of condensate size, reduced numbers of condensates >1 μm², and a smaller average area of total condensates (Fig. 4b–d). Notably, even though the M120I mutation occurs in the N-terminal coiled-coil domain, it had an even stronger impact on droplet formation than the R533* mutation, supporting that N–C termini interaction is involved in condensate formation of CTTNBP2 in crude lysates.

We further analyzed condensate formation of purified ASD-linked CTTNBP2 mutants. All purified GFP-D570Y, -R533* and -M120I mutant proteins generated fewer and smaller condensates than WT (Fig. 4e–h). Moreover, the GFP intensities of these condensates were lower than that of WT (Fig. 4i). Note that D570Y mutant proteins formed bigger condensates in crude lysates (Fig. 4a–d). Potentially, a higher affinity of D570Y mutant protein for cellular components, such as microtubules, offsets its deficits in condensate formation, allowing it to recruit more cellular proteins to form the bigger condensates.

**ASD-linked mutants of CTTNBP2 respond to zinc.** The third set of experiments was to evaluate the effect of zinc on ASD-linked CTTNBP2 mutant proteins in solution. We added zinc at a final concentration of 1, 5, or 10 μM to HEK293T cell lysates and then examined condensate formation. In general, higher zinc concentrations increased the size of condensates, both for WT and mutant proteins (Fig. 5a–c). However, GFP-D570Y aggregates were much larger than GFP-WT aggregates in the presence of 10 μM zinc (Fig. 5a–c), whereas GFP-R533* and GFP-M120I condensates were still smaller than those of GFP-WT at the same zinc concentration (Fig. 5a–c). For purified proteins, zinc treatment also increased the size of all GFP-D570Y, -R533* and -M120I condensates in solution (Fig. 5d–f). These results support that no matter whether WT or ASD-linked CTTNBP2 mutant proteins are being considered, all respond to zinc by forming high-order multimers and aggregates.

Together, these results indicate that all three of the ASD-linked CTTNBP2 mutations we assessed differentially influence the protein's condensate formation ability and mobility. However, none of these three mutations obviously impair the response of CTTNBP2 to zinc (summarized in Fig. S5c).

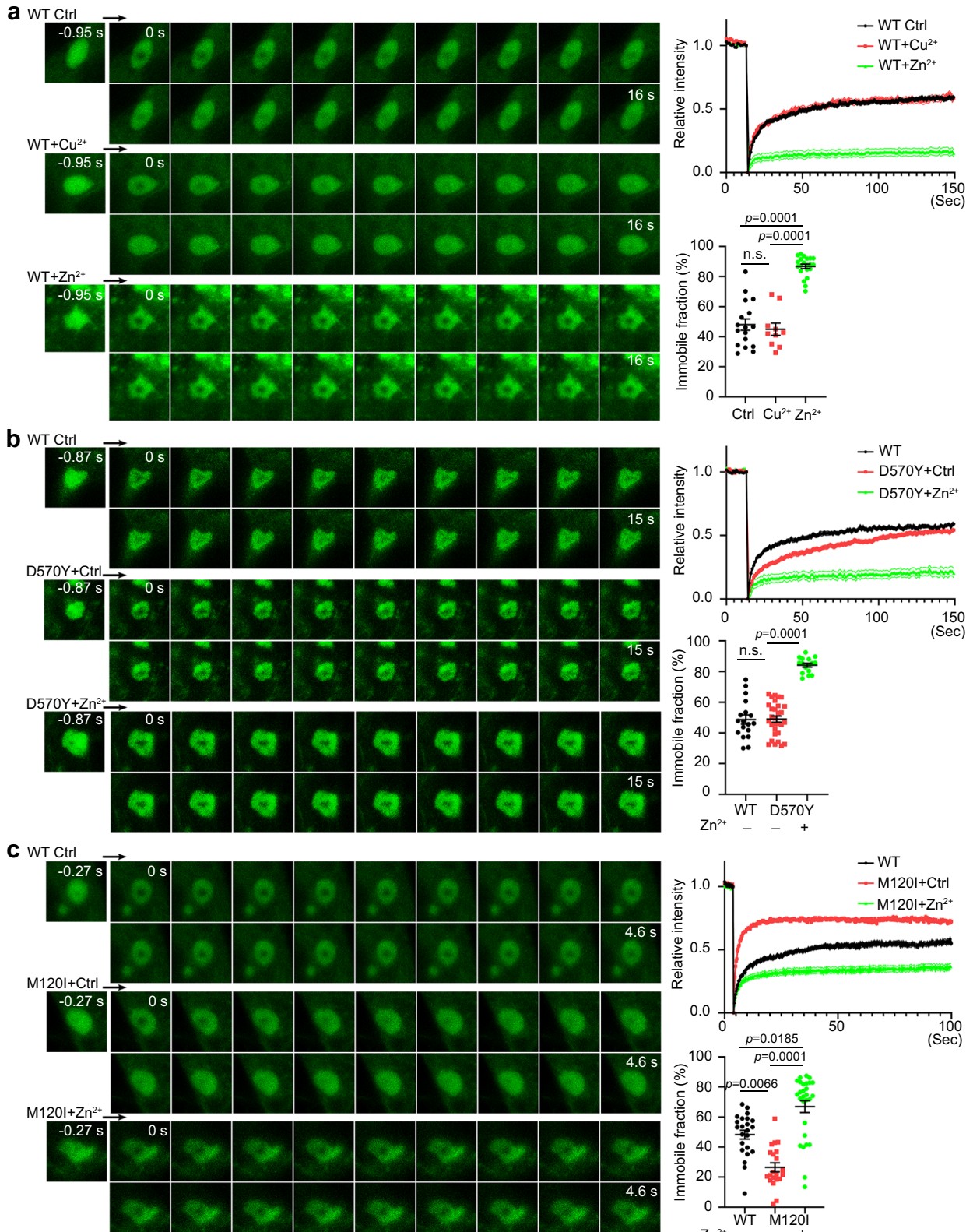

**Fig. 3 Zinc and ASD-linked mutations alter CTTNBP2 mobility in COS1 cells. a** WT GFP-CTTNBP2. **b** GFP-CTTNBP2 D570Y mutant. **c** GFP-CTTNBP2 M120I mutant. FRAP was performed to analyze the protein mobility of GFP-tagged WT and mutant CTTNBP2, as indicated, in COS1 cells. ZnSO$_4$ or CuSO$_4$ (1 mM) was added to culture medium 30 min before photobleaching as indicated to assess their effects. Black arrow indicates the direction of the live-imaging time-series. Normalized fluorescence intensity and the immobile fraction are shown in the plots at right. For immobile fraction plots, a Kruskal–Wallis test with Dunn's multiple comparisons test was conducted for **a** and **c**, and a one-way ANOVA with Bonferroni's multiple comparisons test was conducted for **b**. All statistical analysis and results are summarized in the Statistic Information in the source data file. ns., non-significant. Scale bar: 1 μm.

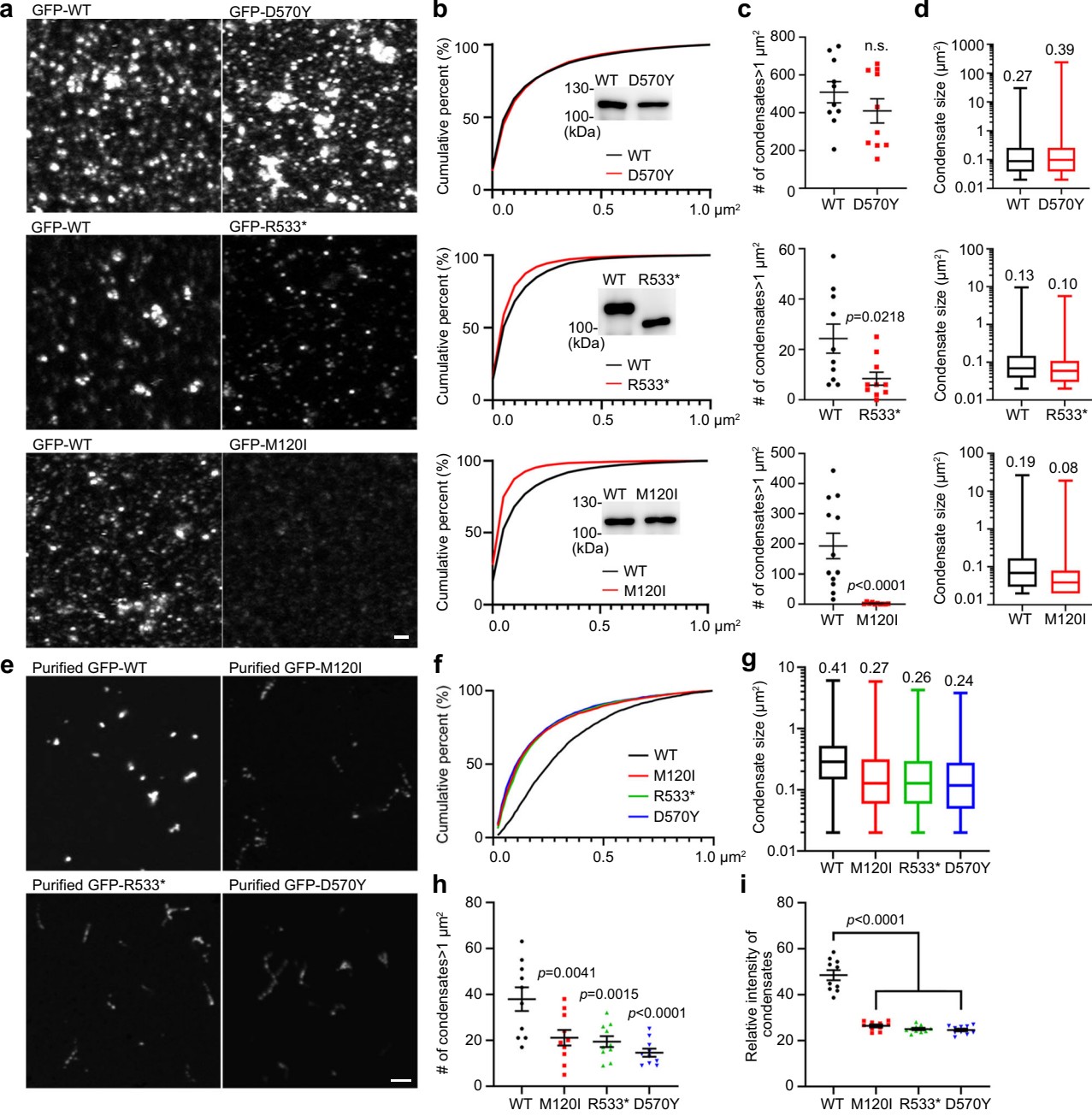

**Fig. 4 ASD-linked CTTNBP2 mutations alter condensate formation. a** Condensate formation by crude lysates of GFP-WT and ASD-linked mutant CTTNBP2. Images of condensates are shown. In different sets of experiments, protein levels varied due to differing cell densities and transfection efficiencies. Condensate numbers of WT proteins may therefore vary in different sets of experiments. To confirm the expression levels of WT and mutant proteins were comparable in the same set of experiments, immunoblotting was performed (shown in **b**, actin loading controls are included in). WT CTTNBP2 was assessed in each set of experiments for comparison with mutant protein. **b** Cumulative probability of CTTNBP2 condensates <1 μm². Expression levels of the mutant and WT constructs were confirmed by immunoblotting, as indicated. **c** Numbers of CTTNBP2 condensates >1 μm². **d** Size of total CTTNBP2 condensates. Box-plots indicate median (middle line), 25th and 75th percentile (box), and min and max (whiskers). The indicated numbers are average sizes. $N = 86801$ (WT, top), 69450 (D570Y), 50488 (WT, middle), 26596 (R533*), 48929 (WT, bottom) and 7763 (M120I) condensates over 10 images collected from two independent experiments. **e** Condensate formation of purified GFP-WT and ASD-linked mutant CTTNBP2. Images of condensates are shown. **f** Cumulative probability of CTTNBP2 condensates <1 μm². **g** Condensate size of total CTTNBP2 condensates. Box-plots indicate median (middle line), 25th and 75th percentile (box), and min and max (whiskers). The indicated numbers are average sizes. $N = 4595$ (WT), 4156 (M120I), 4074 (R533*) and 4088 (D570Y) condensates over 10 images collected from two independent experiments. **h** Numbers of CTTNBP2 condensates >1 μm². **i** Signal intensity of CTTNBP2 condensates in an imaging field. A total of 10 images from each group was randomly selected for quantification. **c** Mann–Whitney test for WT vs. D570Y and WT vs. M120I; two-tailed unpaired *t* test for WT vs. R533*; **h** One-way ANOVA with Bonferroni's multiple comparisons test; i one-way ANOVA with Bonferroni's multiple comparisons test. All statistical analysis and results are summarized in the Statistic Information in the source data file. n.s. non-significant. Scale bars: **a** 1 μm; **e** 2 μm.

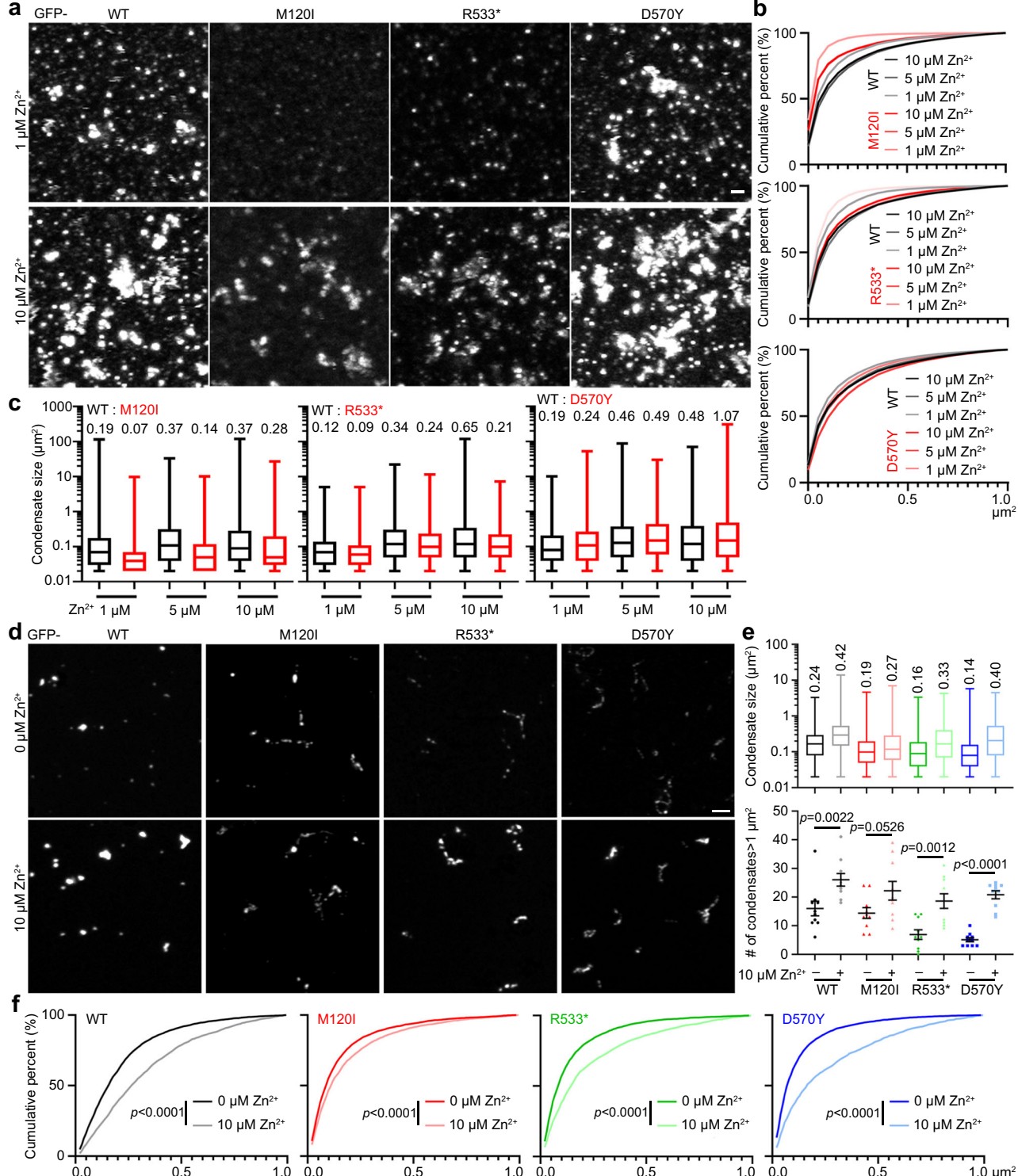

**CTTNBP2 forms condensates in neurons**. Since CTTNBP2 is a neuron-specific protein, we investigated if CTTNBP2 forms condensates in neurons and if that represents a critical function for the protein. Cultured hippocampal neurons are a well-established system for investigating different stages of neuronal development, including axonal and dendritic growth and synaptic formation and maintenance[48]. Using this system, we reported previously that CTTNBP2 forms patches along microtubules at 10 days in vitro (DIV) before dendritic spine formation and that the protein is targeted to dendritic spines when neurons are mature[32–34]. Using

CTTNBP2 antibodies to monitor endogenous CTTNBP2, we also observed that CTTNBP2 formed irregular patches along processes and in the somata of immature cortical neurons at 4 DIV (Fig. 6a). Treatment of cultured neurons with 1,6-HD reduced CTTNBP2 condensation (Fig. 6a–c). We then investigated if CTTNBP2 condensates could reform upon washing 1,6-HD out from cultures. After 1,6-HD washout for 40 or 60 min, the aggregation index of CTTNBP2 increased and differed significantly from the group without washout (Fig. 6d, e), supporting that the effect of 1,6-HD on CTTNBP2 condensates is reversible.

**Fig. 5 $Zn^{2+}$ modifies the condensate (aggregate) formation ability of GFP-WT and ASD-linked mutants of CTTNBP2. a** Representative images of crude lysates showing GFP-WT and ASD-linked mutant CTTNBP2. **b** Cumulative probability of CTTNBP2 condensates $<1\,\mu m^2$ in the presence of different zinc concentrations. **c** Condensate size of total condensates under conditions of various zinc concentrations. Box-plots indicate median (middle line), 25th and 75th percentile (box), and min and max (whiskers). The indicated numbers are average sizes. The numbers (N) of analyzed condensate: left, 44620 (WT $1\,\mu M$), 19227 (M120I $1\,\mu M$), 27351 (WT $5\,\mu M$), 18233 (M120I $5\,\mu M$), 33612 (WT $10\,\mu M$), 7757 (M120I $10\,\mu M$); middle, 36786 (WT $1\,\mu M$), 19423 (R533* $1\,\mu M$), 29687 (WT $5\,\mu M$), 11921 (R533* $5\,\mu M$), 23793 (WT $10\,\mu M$), 20386 (R533* $10\,\mu M$); right, 44463 (WT $1\,\mu M$), 12384 (D570Y $1\,\mu M$), 29876 (WT $5\,\mu M$), 23859 (D570Y $5\,\mu M$), 49531 (WT $10\,\mu M$), 46574 (D570Y $10\,\mu M$) collected from 10 images over two independent experiments. **d** Condensate formation of WT and ASD-linked mutants of CTTNBP2 in the presence or absence of zinc. Representative images of purified GFP-WT and ASD-linked CTTNBP2 mutant proteins are shown. **e** Quantification of **d**. Upper: The size of total condensates. Box-plots indicate median (middle line), 25th and 75th percentile (box), and min and max (whiskers). The indicated numbers are average sizes. Lower: Numbers of CTTNBP2 condensates $>1\,\mu m^2$. **f** Cumulative probability of CTTNBP2 condensates $<1\,\mu m^2$ in the presence or absence of zinc. A total of 10 images from each group was randomly selected for quantification. Lower **e** Mann–Whitney test for WT vs. WT + Zn and D570Y vs. D570Y + Zn; two-tailed unpaired $t$ test for M120I vs. M120I + Zn and R533* vs. R533* + Zn; **f** two-sided Kolmogorov–Smirnov test. All statistical analysis and results are summarized in the Statistic Information in the source data file. n.s., non-significant. Sample sizes in **b**, **c** and upper **e** >1900, so statistical analysis not required. Scale bars: **a** $1\,\mu m$; **d** $2\,\mu m$.

Given that zinc-induced high-order multimerization of CTTNBP2 oligomers is 1,6-HD-resistant (Fig. 2f), the presence of zinc may alter CTTNBP2 condensation upon or after 1,6-HD treatment. To investigate that possibility, zinc was supplemented before adding 1,6-HD and then maintained during recovery. Indeed, we observed that zinc supplementation greatly attenuated the effect of 1,6-HD in terms of dissolving CTTNBP2 condensates in neurons (Fig. 6f, g). Washout of 1,6-HD did not noticeably enhance CTTNBP2 condensation in the presence of zinc (Fig. 6f, g). We also noticed that zinc-treated neurons tended to have bigger CTTNBP2 condensates, as ~25–30% of zinc-treated neurons contained at least one big condensate with the longest axis being >2 µm (Fig. 6f). Those large condensates were rarely found in neurons lacking zinc supplementation (Fig. 6a, d). These results evidence that CTTNBP2 forms dynamic condensates in neurons and that zinc stabilizes CTTNBP2 condensates in immature neurons.

**LLPS and zinc regulate the synaptic distribution of CTTNBP2.** Given that CTTNBP2 is a critical postsynaptic protein for recruiting other proteins to synapses[32,35,36], we examined if LLPS and zinc contribute to the synaptic distribution and synaptic protein–protein interactions of CTTNBP2 in mature neurons. Consistent with previous studies[32,34,36], we observed that endogenous CTTNBP2 formed puncta at dendritic spines of cultured hippocampal neurons at 18–20 DIV (Fig. 7a). Treatment of hippocampal neurons with 1,6-HD reduced CTTNBP2 puncta in dendritic spines, as revealed by linescan analysis and summed enrichment index quantification (Fig. 7b–d). Importantly, the presence of zinc in the cultures enabled more CTTNBP2 to be retained in dendritic spines despite 1,6-HD treatment (Fig. 7a–d), suggesting that CTTNBP2 undergoes LLPS to form condensates in dendritic spines and that zinc enhances synaptic retention of CTTNBP2 in the presence of 1,6-HD.

Next, we investigated if the effect of 1,6-HD is reversable in mature neurons. We conducted dual staining of endogenous CTTNBP2 and F-actin, the cytoskeleton highly enriched in dendritic spines. Directly after 1,6-HD treatment, the size of CTTNBP2 condensates was reduced by 53% compared to the control group without 1,6-HD treatment, but numbers of CTTNBP2 condensates were increased by 17% (Fig. 7e, f). After 1,6-HD washout, cultured neurons were recovered at 20, 40 and 60 min. We found that the size of CTTNBP2 condensates increased relative to the 1,6-HD-treated group but was still smaller than the control group without 1,6-HD treatment (Fig. 7e, f). For F-actin, 1,6-HD treatment reduced aggregate number but not size (Fig. 7e, f). Furthermore, unlike CTTNBP2, both the size and number of F-actin puncta did not increase after 1,6-HD washout (Fig. 7e, f). Thus, although both CTTNBP2 and F-actin

are highly concentrated at dendritic spines, their responses to 1,6-HD treatment differ, implying a specific response of CTTNBP2 to 1,6-HD treatment.

Since synaptic vesicles are the major subcellular compartments for zinc storage in neurons[46] and presynaptic zinc and glutamate are co-released to postsynaptic neurons upon neurotransmission[49], synaptic stimulation is one of the physiological conditions by which postsynaptic zinc levels can be increased. To investigate if synaptic stimulation would also render CTTNBP2 resistant to 1,6-HD, we applied bicuculline to neuronal cultures to enhance neuronal activity at 20 DIV. We found that this treatment indeed attenuated the effect of 1,6-HD in terms of reducing CTTNBP2 at dendritic spines (Fig. 7g–j). Moreover, zinc was involved in the effect of bicuculline because TPEN, a zinc chelator, attenuated the effect of bicuculline on the synaptic distribution of CTTNBP2 (Fig. 7k–m). Thus, neuronal activation increases the stability of CTTNBP2 condensates at synapses.

**LLPS and zinc influence synaptic CTTNBP2–SHANK3 association.** Our previous study evidenced protein–protein interactions between CTTNBP2 and other synaptic proteins[35]. Here, we focused on SHANK3 as an example to further explore the effect of LLPS and zinc on the protein–protein interactions of CTTNBP2. In crude HEK293T lysates, mCherry-SHANK3 formed small condensates in solution (Fig. 8a, upper). When mCherry-SHANK3 was mixed with 4 µM purified GFP-CTTNBP2, these two proteins together formed huge condensates (Fig. 8a; Movie S17).

To further confirm co-condensation of CTTNBP2 with SHANK3, we performed two additional experiments. First, we co-transfected CTTNBP2 and SHANK3 into COS1 cells and performed immunostaining to monitor the distribution patterns of CTTNBP2 and SHANK3. All of the CTTNBP2 condensates in those cells contained SHANK3 (Supplementary Fig. S6), indicating co-condensation of CTTNBP2 and SHANK3 in cellulo. Second, we transfected CTTNBP2 and SHANK3 into HEK293T cells and used the crude lysates to characterize condensate formation. When CTTNBP2 and SHANK3 were singularly transfected into HEK293T cells, they individually formed condensates in crude lysates (Fig. 8b, c). When CTTNBP2 and SHANK3 co-existed, they occurred in the same condensates in crude lysates (Fig. 8b, c). Based on the colocalization coefficient, ~42 ± 3.8% CTTNBP2 formed co-condensates with SHANK3. Compared with CTTNBP2 alone, mixing CTTNBP2 with SHANK3 increased the size of CTTNBP2 condensates (Fig. 8c, left) and vice versa (Fig. 8c, right). Together, these results demonstrate that CTTNBP2 and SHANK3 co-condense, both in solution and in cellulo.

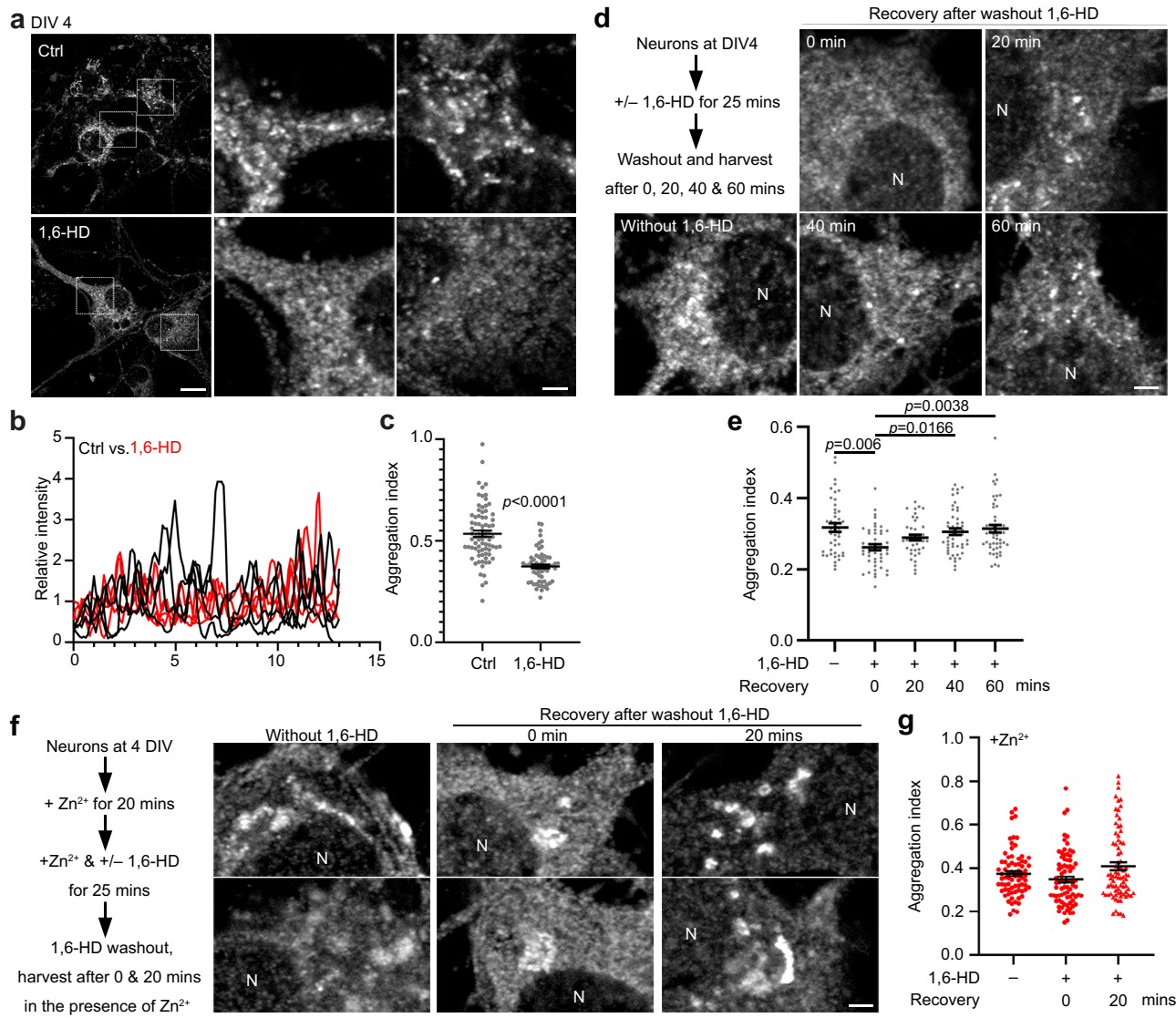

**Fig. 6 CTTNBP2 forms condensates in immature neurons. a** CTTNBP2 condensates in immature hippocampal neurons are sensitive to 1,6-HD treatment. At DIV 4, neurons were treated with 4% 1,6-HD for 25 min and then immunostained with anti-CTTNBP2 antibody. The insets in the low-magnification images are enlarged in the middle and right panels. **b** Linescan quantification of CTTNBP2 immunoreactivity in somata. **c** Aggregation index (see the "Methods" section) of CTTNBP2 immunoreactivity. **d** Experimental procedure to investigate the effect of 1,6-HD washout on reformation of CTTNBP2 condensates. Representative images are shown. **e** Quantification of CTTNBP2 condensate formation and the effect of 1,6-HD washout. **f** Experimental procedure to investigate the effect of zinc on condensate formation of CTTNBP2. Representative images are shown. **g** Quantification of **f**. **c** Mann–Whitney test, two-sided. **e** and **g** two-sided Kruskal–Wallis test with Dunn's multiple comparisons. All statistical analysis and results are summarized in the Statistic Information in the source data file. Scale bars: **a** original, 10 µm; enlarged, 2 µm; **d** and **f** 2 µm.

Next, we investigated the interaction between CTTNBP2 and SHANK3 in mature cultured hippocampal neurons. Both CTTNBP2 and SHANK3 showed punctate synaptic patterns, and they were adjacent to or intermingled with each other (Fig. 8d; Movie S18). Addition of zinc enhanced colocalization of SHANK3 with CTTNBP2 at 18 DIV (Fig. 8e, f), and this effect was abrogated by TPEN treatment (Fig. 8e, f), supporting a specific involvement of zinc. Application of both bicuculline and zinc resulted in an additive effect on CTTNBP2 and SHANK3 colocalization (Fig. 8g), supporting that zinc-related synaptic stimulation impacts the CTTNBP2-SHANK3 association.

To determine if LLPS and zinc are also involved in the synaptic association of CTTNBP2 with SHANK3, we subjected mature neurons at 18 DIV to 1,6-HD treatment and found that synaptic SHANK3 was also dispersed in the same pattern as observed for CTTNBP2 (Fig. 8h–j), with the colocalization coefficient for CTTNBP2 and SHANK3 being reduced to almost zero (Fig. 8j). However, zinc addition enhanced CTTNBP2 and SHANK3 colocalization even in the presence of 1,6-HD (Fig. 8h–j). The reduced colocalization of CTTNBP2 and SHANK3 is unlikely to be due to reduced protein levels because CTTNBP2 and SHANK3 proteins levels were not decreased, as determined by immunoblotting (Fig. S7).

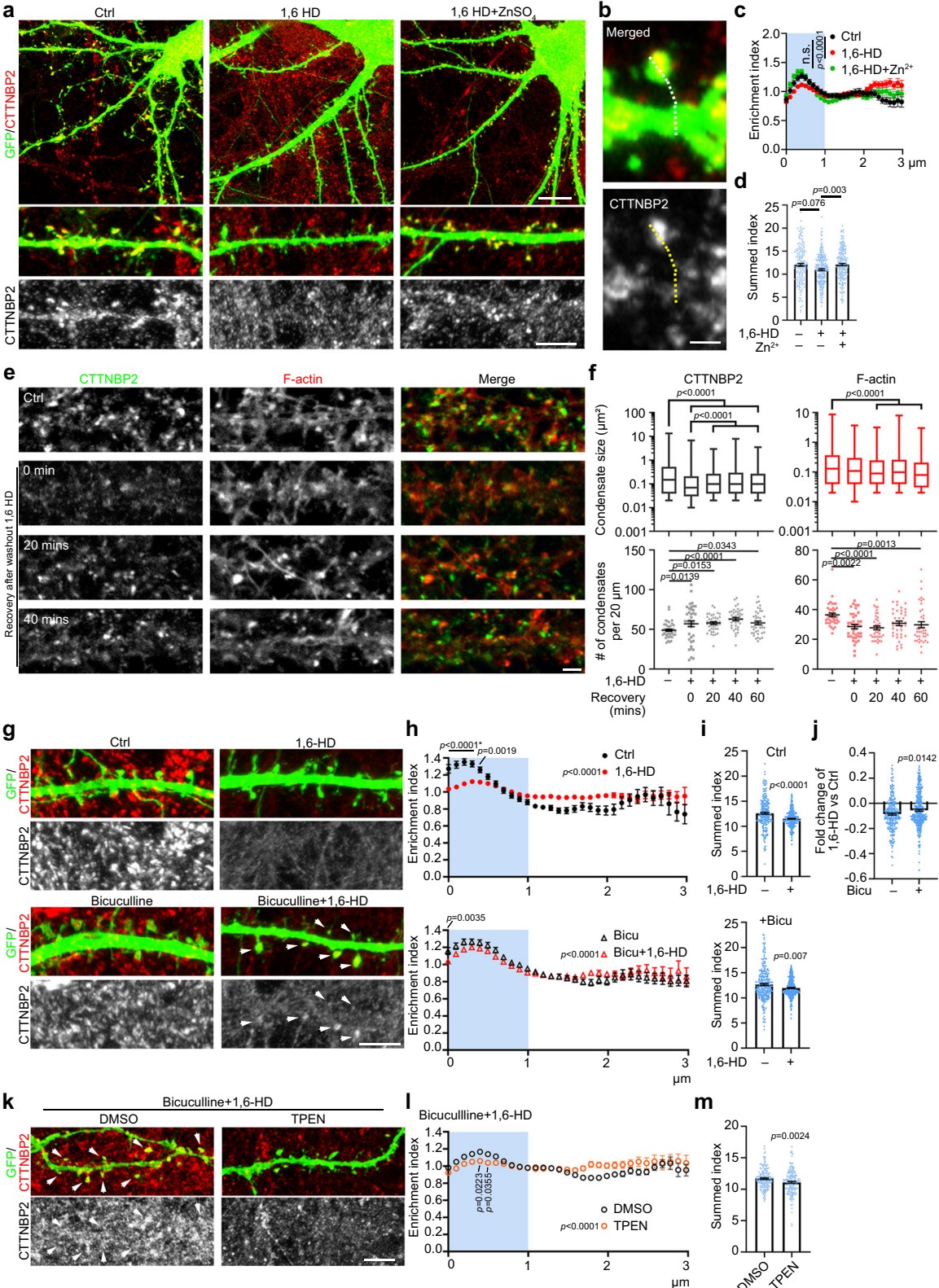

Thus, LLPS is involved in the synaptic distribution of CTTNBP2 and its interaction with SHANK3 in neurons, with synaptic stimulation employing zinc to influence the phase transition and mobility of postsynaptic CTTNBP2 and SHANK3.

**ASD-linked mutations alter synaptic properties of CTTNBP2.** We also investigated the impact of ASD-linked CTTNBP2 mutations on the relevance of LLPS and zinc to the synaptic distribution of CTTNBP2. To mimic monoallelic mutation in patients with *CTTNBP2* deficiency[20] and to avoid the influence of

**Fig. 7 CTTNBP2 synaptic distribution is regulated by phase separation, zinc and neuronal activity. a** CTTNBP2 puncta in dendritic spines are sensitive to 1,6-HD. Representative images are shown. **b** Example of linescan quantification. GFP outlines dendrite morphology and linescan quantification was initiated from the dendritic spine tip to the dendritic shaft. **c** Enrichment index of endogenous CTTNBP2 within dendritic spines under 1,6-HD and zinc treatments. CTTNBP2 fluorescence signals of individual dendritic spines were averaged and calibrated. The enrichment index indicates the relative distribution of CTTNBP2 within dendritic spines. The region within 1 μm of the dendritic spine tip (light blue) represents the spine head. Data are presented as mean values ± SEM. $N = 176$ (Ctrl), 257 (1,6-HD) and 210 (1,6-HD + $Zn^{2+}$) dendritic spines over two independent experiments. **d** 1,6-HD treatment reduces the summed index of endogenous CTTNBP2 within dendritic spines. Relative CTTNBP2 fluorescence signals in spine heads were summed to assess protein enrichment there. **e** Representative images showing the effect of 1,6-HD treatment and recovery after 1,6-HD washout on CTTNBP2 and F-actin aggregations. Mature neurons (18 DIV) were treated as indicated. **f** Quantification of **e**. Size and number of total condensates are shown. Upper: box-plots indicate median (middle line), 25th and 75th percentile (box), and min and max (whiskers); lower: mean and the result of individual neuron. **g** Neuronal activity retains CTTNBP2 in dendritic spines. Neurons were activated via 40-μM bicuculline treatment for 30 min and then treated with 4% 1,6-HD for 15 min. Representative images of endogenous CTTNBP2 and transfected GFP are shown. **h** Enrichment index of endogenous CTTNBP2 within dendritic spines under conditions of 1,6-HD and bicuculline (co)treatment. Data are presented as mean values ± SEM. $N = 269$ (Ctrl), 313 (1,6-HD) and 291 (Bicu) and 323 (Bicu+1,6-HD) dendritic spines over two independent experiments. **i** Summed index of **h**. **j** The difference between the summed index for Ctrl and +Bicu in **i** in terms of 1,6-HD treatment. **k** TPEN attenuates the effect of bicuculline on CTTNBP2 synaptic distribution. **l** Linescan of **k**. $N = 176$ (DMSO) and 181 (TPEN) dendritic spines over two independent experiments. **m** Summed index of **l**. Two-way ANOVA with Bonferroni's multiple comparisons test for **c**, **h** and **l**. One-way ANOVA with Bonferroni's multiple comparisons test or Mann–Whitney test for **d**, **i** and **m**. Kruskal–Wallis test with Dunn's multiple comparisons for **f**. All statistical analysis and results are summarized in the Statistic Information in the source data file. **h** *, the *p* value for all indicated comparisons. Scale bars: **a** upper 10 μm; lower, 5 μm; **b** 1 μm; **e** 2 μm; **g** 5 μm; **k** 5 μm.

early developmental deficits[35,36], we transfected the mutant constructs into cultured hippocampal neurons at 14 DIV and analyzed dendritic spine phenotypes at 18–20 DIV. The typically punctate pattern of CTTNBP2 at dendritic spines was lost for the R533* mutant, instead being evenly distributed along the dendrites and somata of mature cultured neurons (Fig. 9a). These results suggest a critical role for the C-terminal region of CTTNBP2 in synaptic localization. Given that LLPS of CTTNBP2 is impaired by the R533* mutation, the reduced synaptic distribution of the R533* mutant protein is consistent with our speculation that LLPS is involved in the dendritic distribution of CTTNBP2.

We further compared WT CTTNBP2 with the D570Y and M120I mutant proteins. Similar to our results for endogenous CTTNBP2 (Fig. 7), 1,6-HD treatment also reduced overexpressed HA-tagged WT CTTNBP2 in dendritic spines (Fig. 9b). Importantly, HA-tagged D570Y mutant proteins were resistant to 1,6-HD treatment and were retained in dendritic spines (Fig. 9b, c), suggesting that the D570Y mutation facilitates LLPS and forms more stable condensates at dendritic spines.

The response of the M120I mutant to 1,6-HD is the opposite of that displayed by the D570Y mutant. It proved more sensitive than WT CTTNBP2 to 1,6-HD treatment, displaying lower levels than HA-tagged WT CTTNBP2 in dendritic spines in the presence of 1,6-HD (Fig. 9d, e). Addition of supplementary zinc to neuronal culture enhanced CTTNBP2 levels in dendritic spines despite 1,6-HD treatment (Fig. 9d, e), though synaptic retention of M120I mutant proteins was still slightly lower than that of WT CTTNBP2 (Fig. 9e). Together, these results suggest that the M120I mutation destabilizes LLPS of CTTNBP2 in neurons, resulting in higher sensitivity to 1,6-HD. Increasing the zinc level ameliorates the deficits of the M120I mutant.

**Dietary zinc supplementation improves mouse social behaviors.** Finally, we investigated if zinc supplementation ameliorates the behavioral defects previously reported for *Cttnbp2*[+/M120I] mice[36]. We conducted three consecutive reciprocal social interaction (RSI) assays[35] to analyze the effect of zinc supplementation on the same *Cttnbp2*[+/M120I] mice. This assay setup was used previously to characterize the effect of zinc supplementation on *Cttnbp2*[+/-] and *Cttnbp2*[+/R533*] mice. Before the first RSI test, mice drank regular water. After the first RSI test, mice were provided 40 ppm zinc in drinking water for 7 days, which has been shown previously to increase zinc levels in the brain[35]. The

second RSI test was then performed to evaluate the effect of zinc supplementation. After the second RSI test, the mice reverted to drinking regular water for 7 days before undergoing the third RSI test (Fig. 9f). We found that zinc supplementation improved the social behaviors of *Cttnbp2*[+/M120I] mice (Fig. 9f, g). After withdrawal from drinking water, the beneficial effect of zinc supplementation could be maintained in some but not all M120I mutant mice (Fig. 9f, g). Thus, continuous zinc supplementation is required to maintain the rescue effect.

In conclusion, together the above-described results support that ASD-linked mutations of CTTNBP2 impact its phase separation and synaptic distribution, and that zinc supplementation can at least partially ameliorate the synaptic deficits of mutant mice and improve their social impairment.

## Discussion

In this study, we revealed two important molecular properties of CTTNBP2. One is protein phase separation and transition, and the other is zinc binding (Fig. 9h). LLPS via the C-terminal IDR results in CTTNBP2 condensate formation. Zinc binds the N-terminal coiled-coil region to induce high-order multimerization of CTTNBP2 oligomers, which immobilizes CTTNBP2 and triggers a liquid-to-gel phase transition. In mature neurons, LLPS and a zinc-induced phase transition control synaptic retention and the synaptic protein–protein interactions of CTTNBP2. Although ASD-linked mutations of *CTTNBP2* differentially alter condensate formation, the mutant proteins all respond to zinc. Therefore, zinc supplementation ameliorates the impact of synaptic retention of ASD-linked CTTNBP2 mutant proteins and improves the social behaviors of *Cttnbp2* mutant mice. Given that zinc has been implicated as a critical nutritional factor relevant to ASD[50,51], our study also reveals a potential mechanism underlying the role of zinc in synaptic regulation and the relevance of zinc deficiency to sensitivity to ASD-linked genetic variation.

Our finding of an interaction between CTTNBP2 and zinc reveals three notable features of that relationship. First, although it remains elusive how zinc binds CTTNBP2, our results show that the N-terminal coiled-coil domain of CTTNBP2 alone is sufficient for zinc binding. Coiled coils of α-helices are known to chelate zinc[41,42,52]. However, since high zinc concentrations induce CTTNBP2 protein precipitation, it is very difficult to directly prove it using x-ray crystallography or cryo-electron microscopy.

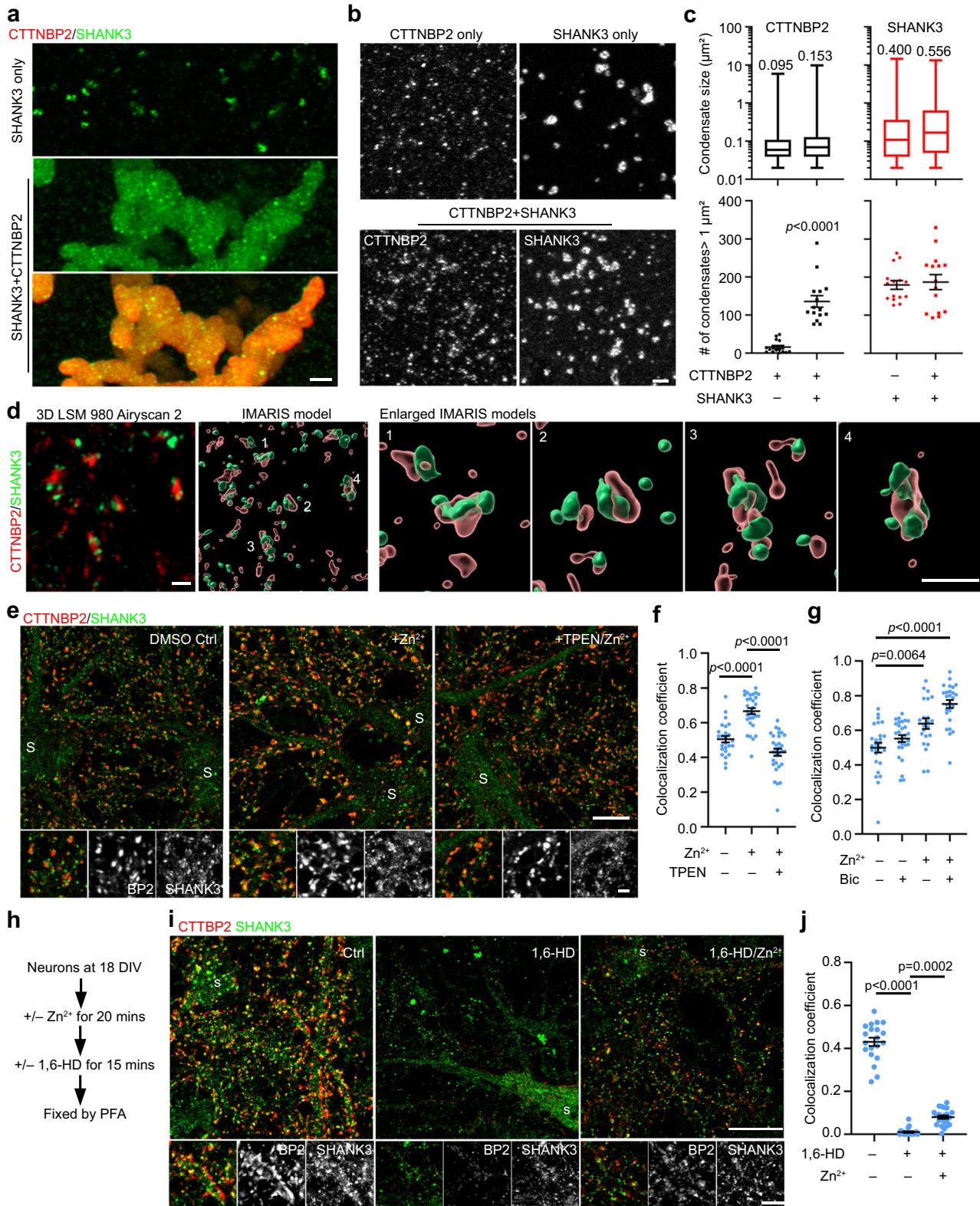

Second, CTTNBP2 displays a relatively low affinity for zinc ($K_d \sim 7\ \mu M$), which is comparable to that of glial-specific S100b protein ($K_d \sim 1\ \mu M$)[53], but much lower than classical zinc finger proteins ($K_d$ values of $10^{-9}–10^{-11}$ or even higher)[54]. Nevertheless, CTTNBP2 may still play a crutial role in the function of zinc in neurons. Given that CTTNBP2 form condensates in neurons, the local concentration of CTTNBP2 in dendritic spines is expected to be much higher than those in our ITC experimental condition and

thereby facilitate its zinc binding in dendritic spines. Moreover, as zinc concentration in synaptic vesicles can be up to 1 mM, high amounts of zinc are co-released with glutamate into the synaptic cleft of glutamatergic synapses during neurotransmission[46,55]. Several postsynaptic channels and transporters—such as voltage-gated L-type $Ca^{++}$ channels, $Na^+/Zn^{++}$ exchangers, N-methyl-D-aspartate (NMDA) receptor, $Ca^{++}$-permeable AMPA/kainate channels, and zinc transporters (such as ZIP4)—then mediate zinc

**Fig. 8 Phase separation and zinc influence the synaptic association of CTTNBP2 with SHANK3. a** Co-condensation of mCherry-SHANK3 (green) and GFP-CTTNBP2 (red). 3D modeling is available in Movie S17. Upper: Crude lysate expressing mCherry-SHANK3 in the presence of PEG, monitored using super-resolution Z-series imaging. Middle and lower: Lysate expressing mCherry-SHANK3 was mixed with 4 µM purified GFP-CTTNBP2 for >20 min. **b** Condensate formation of GFP-CTTNBP2 and mCherry-SHANK3 in crude lysates. Upper: Representative images of GFP-CTTTNBP2 alone or mCherry-SHANK3. Lower: Representative images of GFP-CTTTNBP2 and mCherry-SHANK3 co-condensation. **c** Quantification of **b**, including condensate size and number of condensates > 1 µm$^2$. For condensate size, box-plots indicate median (middle line), 25th and 75th percentile (box), and min and max (whiskers). The indicated numbers are average sizes. **d** Super-resolution imaging using LSM980 with Airyscan 2 and 3D Imaris modeling of endogenous CTTNBP2/ SHANK3 complexes in mature cultured hippocampal neurons at DIV 18. In the 3D Imaris models, semi-transparent CTTNBP2 (red) signals intermingle with or are inserted within SHANK3 condensates (green). Four enlarged images of the CTTNBP2-SHANK3 complexes are shown at right. **e** Addition of zinc enhances CTTNBP2 and SHANK3 colocalization. TPEN, a zinc chelator, reduces it. **f** Quantification of **e**. **g** Bicuculline (40 µM) and zinc exhibit additive effects on CTTNBP2 and SHANK3 colocalization. **h** Experimental procedure for **i**. **i** 1,6-HD treatment disrupts CTTNBP2-SHANK3 colocalization in mature cultured hippocampal neurons at DIV 18. Addition of zinc abrogates the impact of 1,6-HD treatment by retaining some degree of CTTNBP2 and SHANK3 colocalization. **j** Quantification of CTTNBP2–SHANK3 colocalization upon 1,6-HD and zinc treatments shown in **i**. For number of condensates >1 µm$^2$ **c**, two-sided Mann–Whitney test for CTTNBP2 alone vs. CTTNBP2 + SHANK3; two-tailed unpaired *t* test for SHANK3 alone vs. SHANK3 + CTTNBP2. Kruskal-Wallis test with Dunn's multiple comparisons test for **f**, **g** and **j**. All statistical analysis and results are summarized in the Statistic Information in the source data file. Scale bars: **a** original, 5 µm; enlarged, 1 µm; **b** 2 µm; **d** 1 µm; **e** original, 10 µm; enlarged, 2 µm; **i** original, 10 µm; enlarged, 5 µm.

uptake at the postsynaptic site[56–62]. Since CTTNBP2 is highly enriched in dendritic spines, it may bind zinc when the local zinc concentration is greatly augmented at postsynaptic sites upon synaptic stimulation. Then, when zinc levels diminish, CTTNBP2 releases zinc for uptake by high-affinity zinc-binding proteins. Thus, we envisage that CTTNBP2 acts as a zinc sponge to locally store transiently increased zinc levels during synaptic stimulation. In the absence of CTTNBP2, zinc is less efficiently maintained at postsynaptic terminals, which may impair synaptic responses.

Third, our findings also indicate that zinc immobilizes CTTNBP2 at dendritic spines, likely by inducing higher-order multimerization and the liquid-to-gel phase transition of the protein. These data are consistent with our previous finding that CTTNBP2 remains in dendritic spines during activity-induced remodeling, despite its interacting proteins, such as cortactin, striatin, and zinedin, relocating into the dendritic shaft under the same conditions[32,33]. Thus, the interaction of CTTNBP2 with zinc appears to exert two functions; one is to buffer zinc at postsynaptic terminals[63], and the other is to induce the liquid-to-gel phase transition that immobilizes CTTNBP2 in dendritic spines.

Our previous study demonstrated that neuronal activation at the hippocampus and amygdala is impaired in *Cttnbp2* mutant mice[35]. For that reason, here, we used cultured hippocampal neurons to explore the regulatory roles of LLPS and zinc in synaptic targeting and protein interactions of CTTNBP2. Although various ASD-linked CTTNBP2 mutations influence different molecular functions of CTTNBP2, they all manifest as similar morphological defects, i.e. reduced number and size of dendritic spines, in both cultured neurons and hippocampal CA1 neurons in brains[35,36]. Our current study further shows that the M120I, R533* and D570Y mutations all alter LLPS of CTTNBP2, supporting the relevance of CTTNBP2 LLPS to dendritic spine formation and maintenance. More importantly, it further strengthens the significance of synaptopathy in ASD.

ASD-linked genes generally exhibit high scores for intrinsic protein disorder[64], and LLPS is considered highly relevant to various biological processes controlled by certain ASD-linked genes, including chromatin regulation (*AUTS2* and *CHD8*), transcription (*TCF4*), translation (*FMRP*), and RNA splicing (*hnRNPA1* and *RBFOX1*)[5,64]. Some ASD-linked postsynaptic density (PSD) proteins, such as SYNGAP1 and SHANK3, undergo LLPS to form droplet-like condensates with other PSD proteins in solution[5,12,28]. In our study, we investigated CTTNBP2 condensates in neurons and report how LLPS is relevant to synaptic targeting and retention of CTTNBP2,

further strengthening the evidence for a role for LLPS in ASD etiology.

Synapses still undergo dynamic regulation and alterations throughout their entire lifespan. Morphological shrinkage vs. enlargement, maintenance vs. elimination, and activity plasticity continuously impact synapses after they have formed. Thus, how synaptic ASD proteins control synaptic formation and function is not limited to the early developmental stage. Importantly, we previously showed that CTTNBP2 is also critical for the maintenance of dendritic spines[32], supporting a long-lasting impact of CTTNBP2 on synaptic number and size. Certainly, a system, such as a recently published mouse model[65], to monitor neuronal activity/morphology and link it to behavioral change throughout development would prove very useful in future for live-recording the features of ASD-relevant gene mutant mice.

A recent proteomic analysis showed that *Cttnbp2* knockout alters the synaptic distribution of more than 100 proteins[35]. Some CTTNBP2-regulated proteins physically associate with CTTNBP2. The CTTNBP2 condensation we have demonstrated herein reveals a potential mechanism by which CTTNBP2 effectively interacts with and facilitates synaptic targeting of multiple interacting proteins.

In conclusion, we have demonstrated that the IDR-containing synaptic protein CTTNBP2 undergoes LLPS to form condensates, and that zinc binding by CTTNBP2 induces its higher-order multimerization and clustering, thereby enhancing its synaptic retention. ASD-linked mutations in the *CTTNBP2* gene differentially impact LLPS, but all resulting mutant proteins respond to zinc. Our study represents an example of how LLPS and zinc supplementation integrate to control an ASD-linked postsynaptic protein and reveal a mechanism underlying how zinc deficiency may aggravate ASD-linked genetic variation and synaptopathy.

## Methods

**Antibodies, chemicals, and plasmids**. Antibodies, plasmids, and reagents used in this study were as follows: CTTNBP2 plasmids and antiserum 9W (1/500) and purified antibody A5 (0.5 µg/ml);[32,35] HA tag (Cell Signaling, C29F4, 0.5 µg/ml; Roche, 3F10, 0.5 µg/ml); GFP (Abcam, ab13970, 0.5 µg/ml; Invitrogen, A6455, 1/ 1000); β-actin (Sigma-Aldrich, AC-74, 0.5 µg/ml); SHANK3 (Synaptic System, 162 304, antiserum, 1/500); horseradish peroxidase-conjugated secondary antibodies (GE healthcare, NA931, anti-mouse IgG, and NA934, anti-rabbit IgG; Sigma-Aldrich, A7289, anti-guinea pig IgG; 1/5000 dilution for immunoblotting); Alexa Fluor-conjugated secondary antibodies (Invitrogen, A-11039, anti-chicken IgY, Alexa 488; A-21206, anti-rabbit IgG, Alexa 488; A-11073, anti-guinea pig IgG, Alexa 488; A-21424, anti-mouse IgG, Alexa 555; A-21429, anti-mouse IgG, Alexa 555; A-21209, anti-rat IgG, Alexa 594; 1/500 dilution for immunostaining); mCherry-Shank3 (a gift from Dr. H.D. (Harold) Mac Gillavry at University Utrecht, Netherlands); zinc sulfate heptahydrate (Sigma-Aldrich, Z0251-100G); poly(ethylene glycol)-6000 (PEG) (Sigma-Aldrich, 81253-250G); N,N,N',

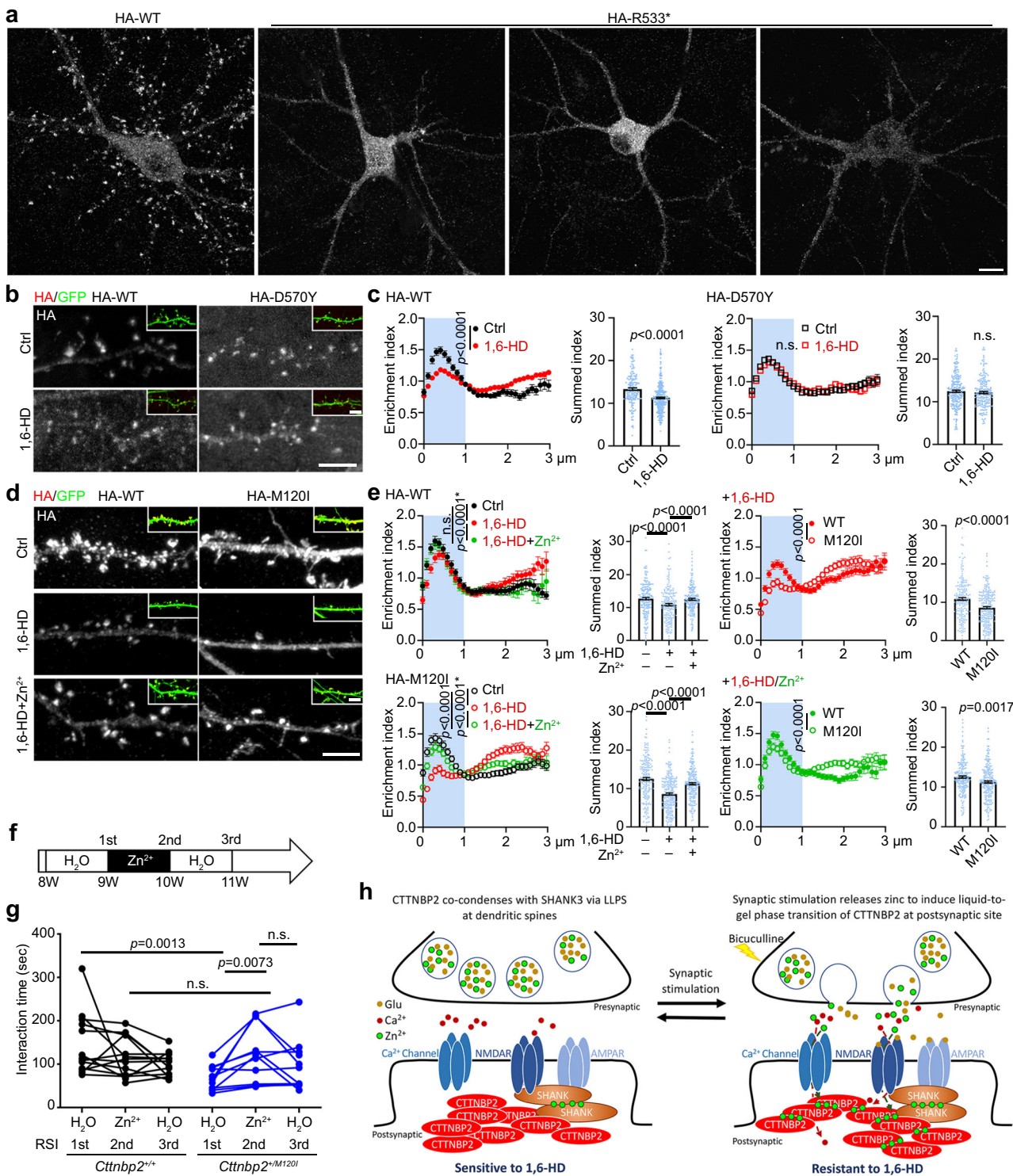

N′-Tetrakis(2-pyridylmethyl)ethylenediamine (TPEN) (Sigma-Aldrich, P4413-50MG); digitonin (Sigma-Aldrich, D141-100MG); 1,6-Hexanediol (Sigma-Aldrich, 240117); GST resin (GE Healthcare, 17-0756-04); NTA agarose (Qiagen, 163023770); Alexa Fluor 546 Phalloidin (Invitrogen, A22283).

**Animals**. The wild-type male and female C57BL/6 mice in this study were purchased from the National Laboratory Animal Center, Taiwan. Cultured hippocampal neurons were prepared from mixed-sex mouse embryos at embryonic days 17-18. *Cttnbp2* M120I mutant mice were generated using a CRISPR/Cas9 editing approach and details have been published previously[36]. M120I mutant mice had been backcrossed to wild-type C57BL/6 mice for more than six generations before experiments. All animal experiments were performed with the approval of the Academia Sinica

Institutional Animal Care and Utilization Committee (Protocol # 12-10-414 and 11-12-294), and in strict accordance with its guidelines and those of the Council of Agriculture Guidebook for the Care and Use of Laboratory Animals, Taiwan. Animals were housed and bred in the animal facility of the Institute of Molecular Biology, Academia Sinica, under controlled humidity (45–55%) and temperature (19–22 °C) and a 12 h light/dark cycle (light off at 20:00). Animals accessed water and food (#5K54, LabDiet) ad libitum. A total of 13 WT and 11 M120I male littermates were used for reciprocal social interaction from 8 to 11 weeks.

**Protein expression and purification**

*GST fusion protein*. CTTNBP2 (amino acids 1–271 and 104–271) and the SAM domain of mouse SHANK3 were amplified by polymerase chain reaction (PCR)

**Fig. 9 ASD-linked mutations differentially impact the synaptic distribution of CTTNBP2 and sensitivity of CTTNBP2 to 1,6-HD treatment. a** HA-tagged R533* mutant protein is evenly distributed in somata and dendritic shafts, but not dendritic spines, of mature neurons at DIV 20. **b** Synaptic HA-tagged D570Y condensates are resistant to 1,6-HD treatment in mature cultured hippocampal neurons (DIV 20). **c** Quantification of **b**, including enrichment index and summed index. **d** Synaptic HA-tagged M120I condensates in mature cultured hippocampal neurons (DIV 20) are susceptible to 1,6-HD treatment, but zinc supplementation can rescue this effect. **e** Quantification of **d**. **f** Procedure for three consecutive reciprocal social interaction tests to evaluate the effect of zinc supplementation on *Cttnbp2+/M120I* mice. **g** The results of **f**. The interaction times of individual *Cttnbp2+/M120I* mice (blue dots linked with blue lines) and WT littermates (black dots linked with black lines) are shown. **h** Model for the roles of zinc and phase separation in the synaptic distribution and protein interactions of CTTNBP2. CTTNBP2 undergoes LLPS to become concentrated in dendritic spines, where it associates with other synaptic proteins such as SHANK3. CTTNBP2 condensates are sensitive to 1,6-HD. When neurons are activated, such as bicuculline treatment, zinc co-released with neurotransmitters from presynaptic vesicles enters the postsynaptic neuron via various ion channels, including voltage-gated calcium channels and NMDARs. Zinc binds to CTTNBP2, promotes the transition of CTTNBP2 from liquid to gel phase, thereby enhancing its synaptic retention and resistance to 1,6-HD. In **c** and **e**, two-way ANOVA with Bonferroni's multiple comparisons test was used to compare enrichment indexes. Kruskal–Wallis tests with Dunn's multiple comparisons test or Mann–Whitney tests were used to compare the summed index within spine heads (light blue areas). In **g**, a two-tailed unpaired *t* test or Mann–Whitney test was used to compare the effect of genetic background. A paired *t* test or Wilcoxon matched-pairs signed-rank test was used to compare the effect of treatments. All statistical analysis and results are summarized in the Statistic Information in the source data file. **e** *, the *p* value for both indicated comparisons. Scale bars: **a** 10 μm; **b** and **d** 5 μm.

and inserted into the pGEX6p1 expression vector (GE Healthcare) containing a PreScission cleavage site. GST-tagged CTTNBP2 and SHANK3 proteins were expressed using the *Escherichia coli* Rosetta strain (Novagen). Expression of fusion proteins was induced by overnight incubation with 0.5 mM IPTG at 18 °C. Bacteria were harvested and resuspended in lysis buffer [20 mM HEPES (pH 7.4), 150 mM NaCl, 3 mM DTT]. The cells were lysed by French press and the lysate was centrifuged for 30 min at 15,000 × *g*. The supernatant was incubated with GST resins (GE Healthcare), and the GST-fused protein was then eluted in buffer containing 50 mM of reduced glutathione. The GST-tag was removed by PreScission protease treatment before protein samples were further purified through MonoQ columns and size-exclusion chromatography (SEC) columns (Superdex 200 16/60). Protein quality was analyzed and confirmed by SDS–PAGE and Coomassie blue staining. All proteins were further concentrated and stored at −80 °C.

*GFP-tagged protein.* We used the Multibac system to generate baculovirus strains expressing GFP-tagged full-length CTTNBP2. Full-length CTTNBP2 was amplified by PCR and cloned into pACEB1 vector (ATG:biosynthetics GmbH). Restriction sites needed for cloning into the MultiBac vectors were included in the PCR primers. GFP and TEV-cleavable ZZ tags were engineered in-frame at the 5′ end of CTTNBP2. The expression construct was sequence-verified and transformed into DH10MultiBac^Turbo cells (ATG:biosynthetics GmbH) to generate bacmids. High-Five insect cells (ThermoFisher) were infected with baculovirus expressing CTTNBP2. Cells were collected at 1000 × *g* for 15 min and resuspended in lysis buffer that contained 50 mM K-phosphate (pH 7.4), 100 mM KCl, 0.1% Tween-20, 5 mM β-mercaptoethanol, 0.1 mM phenylmethylsulphonyl fluoride, and complete EDTA-free protease inhibitors. After sonication, cell lysate was centrifuged for 30 min at 40,000 × *g*. The supernatant was then applied to a lgG-Sepharose 6 fast-flow affinity column (GE Healthcare). Then, the column was equilibrated with washing buffer (50 mM K-phosphate pH 7.4, 100 mM NaCl, 0.1% Tween-20, and 5 mM β-mercaptoethanol). To remove the ZZ tag, TEV protease was loaded onto the equilibrated column and incubated for 3 h at 4 °C. GFP-tagged CTTNBP2 was eluted with washing buffer. Fractions containing the proteins were pooled and further purified over a 10/300 Superdex 200 column. Purified proteins were concentrated and stored in a buffer consisting of 20 mM HEPES pH 7.4, 150 mM NaCl, and 3 mM DTT at −80 °C.

**Circular dichroism**. Recombinant CTTNBP2$_{104-271}$ (0.2 μg/ml) was incubated in binding buffer (20 mM HEPES, 150 mM NaCl, 3 mM DTT pH 7.4) with either different concentrations of ZnCl$_2$ (Fig. 2a) or 1 mM of different metal solutions for 1 h at 4 °C (Fig. 2c), and then centrifuged at 15,700 × *g* for 10 min at 4 °C. The supernatant was subjected to circular dichroism analysis. UV CD spectroscopy was carried out at 25 °C on an AVIV model 400 circular dichroism spectrometer equipped with a temperature-controlled quartz cell of 5 mm pathlength. The spectra shown are averages of three accumulations after filtering, baseline subtraction, signal averaging and digital smoothing using the Aviv CDS software package.

**Zinc-induced pelleting assay**. Recombinant CTTNBP2 (amino acids 104–271) was incubated with different concentrations of ZnCl$_2$ for 1 h at 4 °C. After centrifugation, pellet fractions were dissolved in equal volumes of binding buffer. Both pellet and supernatant fractions were loaded on SDS–PAGE and detected by Coomassie blue staining. Band intensities were analyzed using a BioSpectrum 815 CCD camera and VisionWorx software.

**Zinc pull-down assay**. Recombinant CTTNBP2 (amino acids 1–271, 10 μM) and SHANK-SAM (10 μM) were incubated in 500 μl of binding buffer (50 mM

K-phosphate pH 7.4, 150 mM NaCl, 3 mM ß-mercaptoethanol), containing 50 μl NTA-Agarose (Qiagen) charged with or without Zn$^{2+}$ for 1 h at 4 °C. After a brief centrifugation at 4 °C, unbound and bound fractions were analyzed by SDS–PAGE and Coomassie blue staining. Band intensities were analyzed using a BioSpectrum 815 CCD camera and VisionWorx software.

**Isothermal titration calorimetry (ITC)**. Binding affinities between CTTNBP2 (amino acids 1–271) and zinc chloride were measured using a MicroCal iTC200 system. CTTNBP2 (30 μM) was dialyzed against ITC buffer (20 mM HEPES pH 7.4, 150 mM NaCl, 1 mM β-mercaptoethanol) and stored in the sample cell. ZnCl$_2$ (500 μM) was injected into the cell using a syringe. ITC was performed at 25 °C. The experimental data were fitted to theoretical titration curves using the system software.

**Primary hippocampal neuronal culture and transfection**. The procedures for primary neuronal culture have been described previously[66]. Briefly, mouse hippocampi from embryonic days 17–18 embryos were collected and then washed three times with cold Hank's balanced salt solution (HBSS), before being digested with papain solution (0.6 mg/ml papain, 0.5 mM EDTA, 1.5 mM CaCl$_2$, 0.06% DNaseI, 0.2 mg/ml cysteine) at 37 °C for 15 min. After washing twice with pre-warmed neural maintenance medium [50% Neurobasal medium, 50% DMEM, 1% Pen-Strep (Gibco), 2 mM glutamine (Gibco)], hippocampal neurons were dissociated by gentle pipetting in 10 ml of neural maintenance medium and centrifuged at 100 × *g* for 5 min. After removing the supernatant medium, the cells were resuspended in fresh pre-warmed neural maintenance medium and seeded on poly-l-lysine (1 mg/ml)-coated glass coverslips or culture dishes.

Transfection was performed at 14 days in vitro (DIV) by using a calcium phosphate precipitation method[34,35,66]. Neurons were pre-incubated with 0.5 ml of transfection medium (10% neurobasal medium, 90% DMEM, 0.4% B27 supplement, and 0.1% glutamine) in a 10% CO$_2$ incubator at 37 °C. To prepare DNA precipitates (for each well in 12-well dishes), 50 μl 0.25 M CaCl$_2$ solution containing 5 μg of plasmid DNA was added in a dropwise manner into 50 μl of 2× HeBS solution (274 mM NaCl, 10 mM KCl, 1.4 mM Na$_2$HPO$_4$, 15 mM glucose, 42 mM HEPES pH 7.05–7.07) with mild vortexing. The mixtures were incubated at room temperature for 15 min and added in a dropwise manner into neuronal cultures pre-incubated with transfection medium. Neurons were then incubated in a 5% CO$_2$ incubator at 37 °C for 60 min, before being washed with wash buffer (135 mM NaCl, 20 mM HEPES, 4 mM KCl, 1 mM Na$_2$HPO$_4$, 2 mM CaCl$_2$, 1 mM MgCl$_2$, 10 mM glucose pH 7.3) and DMEM. Both wash buffer and DMEM were pre-incubated for more than 30 min at 37 °C in a 10% CO$_2$ incubator to adjust the pH. Finally, neurons were transferred back to original neural maintenance medium.

**Condensation assay using HEK293T cells and purified protein**
*Crude HEK293T lysates.* To express GFP-tagged CTTNBP2 in HEK293T cells, plasmids were transfected using PolyJet (Signagen) based on the manufacturer's instructions. In brief, for each well of 6-well plate, 1 μg of plasmid DNA in 50 μl DMEM was mixed with 3 μl of PolyJet in 50 μl DMEM. For each 60-mm dish, 2.5 μg of plasmid DNA in 100 μl DMEM was mixed with 7.5 μl of PolyJet in 100 μl DMEM. The mixtures were incubated at room temperature for 15 min and then gently added to the culture. Two days after transfection, cells were harvested in saline (100 mM Tris–Cl, 0.8% NaCl, pH 7.4) containing protease inhibitors and grinded with a dounce homogenizer using a loose pestle. Cell debris and nuclei were removed by centrifugation at 15,700 × *g* for 10 min at 4 °C. Final reaction mixtures containing 47.5 μl protein extract and 1.5% PEG-6000, with or without other treatments as described in the text, were loaded onto glass slides with single

cavities (Assistent) and covered by a glass coverslip for imaging. Images were processed in ImageJ to quantify condensate number and size using the "analyze particle" function. Signal noise was filtered using the same parameter settings among images from the same batch of experiments and the images were then transformed into binary signals for quantification.

*Purified GFP-tagged proteins.* Purified GFP-tagged CTTNBP2 proteins (80–400 nM) were diluted with HEPES buffered saline (20 mM HEPES pH 7.4, 150 mM NaCl and 3 mM DTT) with 3% of PEG-6000 for condensate formation. To investigate the effect of zinc on CTTNBP2 phase transition, we added 1, 5, 10 or 100 μM ZnSO₄ into the protein mixtures. All protein solutions were incubated at room temperature for at least 10 min before monitoring by microscopy. To examine the effect of zinc withdrawal on maintenance of the CTTNBP2 gel phase, after being incubated with 10 μM ZnSO₄ for 30 min, TPEN (15 μM) was added into the mixture and incubated for another 30 min. The solution was then monitored by microscopy. Quantification was performed as above for crude HEK293T lysate.

**Immunostaining.** Before fixation, cells were washed with PBS to remove remaining culture medium. Cells were fixed with 4% paraformaldehyde in PBS for 10 min and permeabilized with 0.2% Triton-X100 in PBS for 10 min at room temperature (RT). Cells were then blocked with 10% BSA in PBS for 30 min at RT, and hybridized overnight with primary antibodies in 3% BSA in PBS at 4 °C. After washing out the unbound antibodies, secondary antibodies or phalloidin-546 (1:100) in 3% BSA in PBS were applied at RT for 1 h. After washout, mounting medium [1% DABCO in PBS] and coverslips were applied for imaging using a confocal LSM 700 or 980 microscope with Airyscan 2 technology (Carl Zeiss).

**Fluorescence recovery after photobleaching (FRAP).** FRAP experiments were performed using an inverted ZEISS 980 confocal microscope (Carl ZEISS) with a Plan-Apochromat ×40/1.3 NA oil-immersion objective. Cells were kept at 37 °C and allowed equilibrate for 30 min before data acquisition. Fifteen image scans were conducted, followed by a bleach pulse of 5 ms on a spot of one pixel within a reference area. Bleaching was performed using a 10-mW 488 nm laser operating at 80% laser power. A series of 385 single-focal-plane images were collected at low laser intensity (0.02% of the 10 mW laser), with 270-ms intervals for the M120I mutant and 860-ms intervals for other groups. Image size was 926 × 263 pixels and the pixel width was 92 nm. To quantify the dynamic fluorescence signals, a region outside of the cell labeled as "background" and a region with fluorescent signal but not affected by photobleaching labeled as "reference" were first identified to correct the data at each time-point for any bleaching artifacts that occurred during the imaging process. Mean fluorescence intensities within the regions were used. Mobile and immobile fractions within selected bleached regions were then quantified. All images were processed in a blinded manner using the FRAP function in ZEN (black edition) with a mono-exponential model for data-fitting.

### Protein distribution in cultured hippocampal neurons

*Immature neurons.* To study the effect of 1,6-HD on CTTNBP2 in immature neurons, cultured hippocampal neurons at 4 DIV were treated with 4% 1,6-HD and 5 μg/ml digitonin for 25 min (Fig. 6a), followed by 1,6-HD washout from culture for recovery for 0, 20, 40, or 60 min (Fig. 6d), before being fixed with 4% paraformaldehyde for immunostaining. For 1,6-HD washout, the residual solution on coverslips was removed as completely as possible. The coverslips were then rinsed with medium once before being placed back in the culture medium for recovery for 0, 20, 40 or 60 min. To examine the effect of zinc, 0.9 mM zinc was added in culture medium before, during and after 1,6-HD treatment (Fig. 6f). To quantify CTTNBP2 aggregation levels, we randomly performed linescanning across the cytoplasm but avoided the nucleus. The signal intensity of CTTNBP2 along the lines was measured and averaged. The values of individual peaks were then normalized against average intensity. The differences between normalized peak values and averages were determined to indicate levels of aggregate formation.

*Mature neurons.* To study the effect of 1,6-HD on CTTNBP2 in mature neurons, cells were incubated with culture medium containing 4% 1,6-HD and 5 μg/ml digitonin for 15 min, before being fixed for staining. To study reformation of CTTNBP2 condensates in mature neurons, cells were treated with 4% 1,6-HD and 5 μg/ml digitonin for 20 min, subjected to 1,6-HD washout, and then incubated in growth medium for 0, 20, 40, or 60 min, before being fixed for staining. To study the effect of zinc on 1,6-HD-induced CTTNBP2 redistribution, culture medium was pretreated with 0.9 mM ZnSO₄ for 20 min before undergoing 1,6-HD treatment (Fig. 7a). To study the effect of neuronal activity on 1,6-HD-induced CTTNBP2 redistribution, neurons were supplied with 15 μM ZnSO₄ in culture medium at DIV 18. At DIV 20, neurons were pre-treated with 40 μM bicuculline in culture medium for 30 min and then treated with 1,6-HD as described above. To confirm the involvement of zinc in the bicuculline-induced redistribution of CTTNBP2, 15 μM TPEN was added during bicuculline treatment. After treatment, the neurons were fixed and stained for microscope-based imaging.

To quantify CTTNBP2 distribution and morphology in the dendritic spines of mature neurons, 20 μm of dendritic segments taken 20 μm away from the soma were selected for quantification. Linescanning from the tip of the dendritic spine to the dendritic shaft (Fig. 7b) was performed in ImageJ[32,67]. Raw fluorescence signal for each scan point along the lines was normalized against the mean intensity of individual lines to acquire an "enrichment index". We define the region within 1 μm of the tip of the dendritic spine as the spine head. To quantify the level of synaptic enrichment within spine heads, we summed the enrichment indexes from spine heads, as shown in Fig. 7b. To assess reduced enrichment (shown in Fig. 7j), the difference of the summed index between control and treatments (Ctrl vs. 1,6-HD or bicuculline vs. bicuculline + 1,6-HD) was normalized against the mean of the summed index from control samples, with a negative value representing a reduction in percentage enrichment.

To study the effect of zinc on localization of CTTNBP2 and SHANK3 in mature neurons, neurons at 18 DIV were treated with 0.9 mM ZnSO₄, in the absence or presence of 0.9 mM TPEN, in culture medium for 20 min and then fixed for staining. The data was directly assessed in Zen software without manual processing to quantify the colocalization coefficient. To model the protein complexes of endogenous CTTNBP2, SHANK3, hippocampal neurons at 18 DIV were fixed and stained with anti-CTTNBP2 and anti-SHANK3 antibodies. Images were captured using a 3D LSM980/Airyscan 2 system in super-resolution mode. The 3D images were processed in Imaris, and modeling was based on fluorescence signals and the manufacturer's instructions.

**Immunoblotting.** To determine the protein levels of wild-type and mutant CTTNBP2 by means of a condensate formation assay, 20 μl protein extract from HEK293T cells was added to 20 μl 2× sample buffer and boiled for 10 min. Each sample (5 μl) was loaded into separate lanes for SDS–PAGE. To validate the protein identity of purified GFP-tagged full-length CTTNBP2, 50 ng of purified protein was also loaded for SDS–PAGE. After protein samples were separated by SDS–PAGE, they were then transferred to PVDF membrane. Membranes were blocked with blocking buffer (5% skimmed milk and 0.1% Tween-20 in PBS) for 30 min and probed with primary antibodies. In principle, primary antibody was added into blocking buffer and hybridized with the membrane overnight at 4 °C. Horseradish peroxidase-conjugated secondary antibodies were used to detect primary antibodies. The results were visualized using WesternBright ECL Spray or Immobilon Western Chemiluminescent HRP Substrate. The original images of full-size gels are shown in the Source Data File.

**Reciprocal social interaction (RSI).** The method for conducting RSI has been reported in detail previously[35]. The test animals were isolated for a week before undergoing RSI. Before interaction, the lid of the test mouse home cage was opened for the entire session to minimize aggressive behaviors. An unfamiliar adult male mouse ~2 weeks younger or of the same age as the test animal was put into the home cage for 10 min. Aggressive behaviors were excluded from the analysis. Mouse behaviors were videotaped from above. To quantify behaviors, the time the test mouse spent sniffing the unfamiliar mouse was manually recorded by an observer. To avoid bias, the animals were relabeled before experiments.

**Zinc supplementation for behavioral assay.** The methodology for zinc supplementation has been reported previously[35]. In brief, to increase zinc intake by mice from 84 to 150 ppm[68], ZnSO₄ was added to drinking water at a concentration of 40 ppm, which was based on daily consumption of ~5 ml water and 2.5 g diet by each mouse. To study the effect of zinc supplementation on the social behavior of *Cttnbp2* M120I mice, three consecutive trials of RSI were performed at intervals of 7 days (Fig. 9f). Test mice were provided with normal drinking water before the first trial, and then switched to zinc-supplemented drinking water for 7 days. After the second trial, the zinc-supplemented drinking water was replaced with normal drinking water for 7 days, before mice underwent the third trial.

**Statistics and reproducibility.** Neuronal morphology and mouse behavior studies were conducted blind to avoid personal bias during quantifications. All experiments were repeated, at least twice, usually three times or more to conclude the reproducibility and statistical analysis. The actual sample sizes of all experiments are summarized in the Statistic Information in the source data file. All image measurements, including staining and immunoblotting analyses, were carried out using ImageJ and Zen software. Statistical analysis and graphical outputs were performed in PRISM v5.03, 8.3 or 9.3.1 (Graphpad software). All data have been tested for normality (D'Agostino and Shapiro–Wilk tests), unless the sample sizes were unsuitable, and outliers were excluded by PRISM. To compare two independent groups of data, a two-tailed unpaired *t*-test was used for normally distributed data or a two-tailed Mann–Whitney test was applied for nonparametrically distributed data. To compare treatment effects (H₂O versus Zn²⁺) on the same animals, two-tailed paired *t*-tests were used for normally distributed data or two-tailed Wilcoxon matched-pairs signed rank test was applied for nonparametrically distributed data. To compare multiple groups of data, one-way ANOVA with Dunnett's or Bonferroni multiple comparison test was used for normally distributed data or a Kruskal–Wallis test with Dunn's multiple comparison test was applied for nonparametrically distributed data. To compare two variants within one dataset, two-way ANOVA with Bonferroni multiple comparison test was used. For all comparisons, $P < 0.05$ was considered significant. Data are mean ± SEM and individual data point. Box-plots indicate median (middle

line), 25th and 75th percentile (box), and min and max (whiskers). Methods and results of all statistical analyses are summarized in the Statistic Information in the source data file. All full scan blots are available in the Source Data file.

**Reporting summary**. Further information on research design is available in the Nature Research Reporting Summary linked to this article.

## Data availability

All experimental data used in this study are available within the Supplementary Movies and Source Data file. All supplementary files and Source Data are provided with this paper.

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

## Acknowledgements
We thank Dr. H.D. (Harold) Mac Gillavry for mCherry-Shank3, and the Imaging Core and Animal Facility of the Institute of Molecular Biology, Academia Sinica, for excellent technical assistance. Dr. John O'Brien conducted English editing and members of Y.-P.H.'s laboratory relabeled samples for blind experiments. This work was supported by grants from the Institute of Molecular Biology, Academia Sinica (to Y.-P.H and K.-C.H.), Academia Sinica (AS-IA-106-L04 and AS-TP-110-L10 to Y.-P.H.), and the Ministry of Science and Technology (MOST 110-2311-B-001-010-MY3 to Y.-P.H.). P.-Y.S. was supported by an Academia Sinica Postdoctoral Fellowship.

## Author contributions
Conceptualization, P.-Y.S., K.-C.H., T.-F.W., and Y.-P.H.; Methodology and investigation, P.-Y.S., Y.-L.F., S.S., S.-P.L., H.C., H.-T.H., K.-C.H., and T-F.W.; Writing, P.-Y.S., Y.-L.F., S.S., K.-C.H., and Y.-P.H.; Funding acquisition, K.-C.H. and Y.-P.H.; Project administration, Y.-P.H. P.-Y.S., and Y.-L.F. contributed equally to the work. K.-C.H. and Y.-P.H. jointly supervised the project. All authors contributed to the article and approved the submitted version.

## Competing interests
The authors declare no competing interests.
