## [Peer Review File · Nature Communications]

REVIEWER COMMENTS

Reviewer #1 (Remarks to the Author):

The manuscript written by Shih et al. investigated the function of cortactin-binding protein 2 (CTTNBP2) in the synaptic formation and modification. The study also discussed a linkage of CTTNBP2 function to neuronal mal-formation associated with autism-like neurodevelopmental disorders. The manuscript is well-written and focused on investigating a particular molecular mechanism with a set of standard, but highly established technical protocols. However, as shown in several cell-function focused studies, a logical gap between in-vitro cellular mechanism and in-vivo human pathological mechanisms is presented.

In this study, the mechanism of CTTNBP2 in the regulation of synaptic interaction in 'current' Petri dish environment was demonstrated. These synapses were collected from mouse hippocampus at some ages of mice (depending on used cell lines). However, the autism is a neurodevelopmental disorder and its neuronal dysregulation/malformation have been implemented from embryo through young adults, in everywhere in the brain and nerve system. Furthermore, the major symptoms expressed and defined as autism are behavioral impairments, which are regulated by the malformed neural system that are already achieved when we find these symptoms. Therefore, to enhance the significance of this study, certain statement regarding how the function/mechanism of CTTNBP2 is involved in the formation processes of autism pathology and the neural networks underlying the regulation of autism symptoms, should be provided in the introduction and the bottom of the discussion.

Specific point-to-point comments:

In the introduction, a topic of SARS-CoV-2 is irrelevant.

Page 4, line 16 from bottom, a citation is needed.

Page 5, line 2, a definition or explanation of IDP should be stated.

Page 6, line 13, It is mentioned that the results from Figure 1b and 1c indicate the formation of membraneless condensates. However, to prove it, evidence supporting the absence of membrane should be presented.

Pages 5-6. Two different cell lines (i.e., Cos-1 and HEK293T) used here need to add explanation about general characteristics of CTTNBP2 proteins on these cell lines. It would be required to add another set of photos of condensate formation in HEK293T to Figure 1b/1c.

Page 6, lines 25-26. Further explanation is required why the formation of spherical droplets in the presence of PEG support the "intrinsic" property of CTTNBP2 in relation to the condensate formation.

Page 20, Fig 2b. Supernatant (S) and Pellet (P) fraction should be labeled in the figure.

Figure 8f. Behavioral data. The data looks messed up here and difficult to understand what occurs here. Need to remake the figure and analysis.

Reviewer #2 (Remarks to the Author):

In this study, authors examine the role of Cortactin-binding protein 2 (CTTNBP2) in dendritic spines. They find that purified CTTNBP2 undergoes PEG-dependent liquid-liquid phase separation (LLPS) via its intrinsically disordered C-terminal domain. CTTNBP2 also binds zinc using its N-terminal coiled-coil domain. Zn binding induces CTTNBP2 aggregation in vitro and in cells. They investigate the effect of Zn and 1,6-hexanediol on synaptic activation and synaptic retention of CTTNBP2 and SHANK3. They also examine the role of

several Autism spectrum disorder mutations on CTTNBP2 phase separation and Zn dependent aggregation. The authors' data that CTTNBP2 binds Zn to regulate its higher order oligomerization and regulate its function is strong and compelling, but the evidence supporting a role for CTTNBP2 LLPS in neurons is lacking.

Although the authors show that purified CTTNBP2 can undergo LLPS in vitro, the cellular experiments are not rigorous enough to conclude CTTNBP2 LLPS is important to the formation or function of CTTNBP2 aggregates in neurons. The main experiments to test for CTTNBP2 LLPS in neurons rely on 1,6-hexanediol, which could potentially cause many off-target effects or just broadly disrupt the PSD. The authors treat cells with 4-8% hexanediol for 25 min. With these conditions, many cellular condensates (including the entire postsynaptic density) could be dissolved. Furthermore, 1,6-hexanediol has been shown to have effects independent of LLPS, such as broadly inhibiting kinase and phosphatase activity (<https://doi.org/10.1016/j.jbc.2021.100260>). Finally, if treatment with 1,6-hexanediol was simply preventing CTTNBP2 LLPS, then the effect of hexanediol should be fully reversible, and CTTNBP2 condensates would reform rapidly after hexanediol washout. In Fig. 5e the authors show that hexanediol treatment in neurons is not reversible, which is not consistent with effects due to LLPS of CTTNBP2. With all of these concerns, it is troubling that the authors rely exclusively on sensitivity to 1,6-hexanediol as evidence for CTTNBP2 LLPS in neurons. Additional experiments, such as expressing CTTNBP2 mutations or truncations that specifically do not undergo LLPS would provide stronger evidence that CTTNBP2 LLPS is important in neurons.

Their data supports the model that Zinc promotes higher-order CTTNBP2 oligomerization, which drives the formation of solid aggregates. They claim that "LLPS is the mechanism by which CTTNBP2 condenses in dendritic spines of cultured hippocampal neurons", but their data does not definitively support this conclusion. Their data more strongly supports the model that Zn-dependent oligomerization and aggregation of CTTNBP2 occurs independently of CTTNBP2 LLPS. The authors show that Zn induced aggregate occurs in the absence of PEG (Fig. S2c). However, LLPS requires PEG (Fig. 1i). These observations suggest that Zn induced CTTNBP2 aggregation does not depend upon CTTNBP2 LLPS in vitro (i.e. Zn induced aggregation does not depend upon a prior LLPS, but can occur de novo as a distinct oligomerization pathway). Similarly, residues 104-271 are sufficient for Zn-dependent oligomerization in vitro (Fig. 2), but a similar fragment (1-271) does not undergo LLPS in vitro (Fig. 1). This also suggests that oligomerization occurs independently to LLPS, not following LLPS. Their data about Zn-dependent oligomerization and aggregation in vitro and in neurons is novel and would be interesting without the need to invoke LLPS.

I also had additional questions/comments about specific figures:

They often show a zoomed up image of a single condensate or small group of condensates from cellular experiments (e.g. Fig 1c, Fig 3). In these figures, could the authors also include an image of the entire cell and highlight the ROI containing the condensates in the zoomed images? It is hard to understand these images without the context of the cell. For example, 1c looks very different than 1b, so it would be helpful to see how these condensates appear in the whole cell.

In Fig. 2b S and P are not labeled in the figure.

Do authors limit or measure the amount of overexpression? The two phenotypes they see in Fig. S3b could be explained by different expression levels. Often one can tell from integrated GFP intensities if cells are expressing lower or higher levels of protein. Since LLPS is highly concentration dependent, different cytoplasmic levels of overexpressed

proteins could lead to different phenotypes.

In Fig. 4, the authors claim that ASD mutations impact phase separation of CTTNBP2. However, these experiments are done in crude cell lysates, thus one cannot rule out the possibility that these mutations are changing interactions with unknown proteins in the lysate. Experiments with purified protein would be required to conclude that these mutations directly change the LLPS propensity of the protein.

The conclude that Shank3 and co-condense. Co-condensing would mean the proteins undergo phase separation together, but the authors are only quantifying co-localization.

Fig. 8f is missing critical information in the figure legend, so it is not possible to understand the data. The black lines and blue lines are two different conditions, but I can't tell what they are from the figure information. Please label the figure appropriately.

Does supplementing Zn in the diet actually change the amount of Zn in neurons or brain tissue? The amount of Zinc in the diet only doubles (84 ppm to 150 ppm) and it's not clear that that would necessarily increase the amount of Zn in neurons. Have the authors or other studies measured the effect of Zn dietary supplements on brain Zn levels?

What happens to Zn-induced aggregates once Zn is removed? This seems to be an important question for their model in figure 8. Zn-induced CTTNBP2 aggregates are solid like and quite stable, so it's possible their turnover would be very slow. Is Zn required to maintain oligomers over time, or is Zn only required for their initial nucleation? If Zn is removed do the aggregates immediately go away, or are they stable over longer periods of time? The authors could assess this in COS cells by treating cells with Zn and then washing out the Zn. If aggregates remained stable after Zn is removed, this could be an interesting mechanism for molecular memory of high Zn.

OUR POINT-BY-POINT RESPONSE TO REVIEWERS

Reviewer #1's general comments: *The manuscript written by Shih et al. investigated the function of cortactin-binding protein 2 (CTTNBP2) in the synaptic formation and modification. The study also discussed a linkage of CTTNBP2 function to neuronal mal-formation associated with autism-like neurodevelopmental disorders. The manuscript is well-written and focused on investigating a particular molecular mechanism with a set of standard, but highly established technical protocols. However, as shown in several cell-function focused studies, a logical gap between in-vitro cellular mechanism and in-vivo human pathological mechanisms is presented. In this study, the mechanism of CTTNBP2 in the regulation of synaptic interaction in 'current' Petri dish environment was demonstrated. These synapses were collected from mouse hippocampus at some ages of mice (depending on used cell lines). However, the autism is a neurodevelopmental disorder and its neuronal dysregulation/malformation have been implemented from embryo through young adults, in everywhere in the brain and nerve system. Furthermore, the major symptoms expressed and defined as autism are behavioral impairments, which are regulated by the malformed neural system that are already achieved when we find these symptoms. Therefore, to enhance the significance of this study, certain statement regarding how the function/mechanism of CTTNBP2 is involved in the formation processes of autism pathology and the neural networks underlying the regulation of autism symptoms, should be provided in the introduction and the bottom of the discussion.*

Our response: First of all, we would like to thank Reviewer #1 for her/his strong support. We agree with Reviewer #1 that we focused on the molecular properties of CTTNBP2 in the original manuscript and did not elaborate on the impact of CTTNBP2 on autism pathology, even though we did employ three autism-linked CTTNBP2 mutations in our study. We very much appreciate Reviewer #1's suggestions. Accordingly, we have extensively rewritten our Introduction and Discussion to emphasize the relevance of LLPS and zinc binding in CTTNBP2-related autism spectrum disorder (ASD) etiology. The altered parts of our manuscript are highlighted in blue font. Below, we summarize some points for Reviewer #1's information.

Neuronal culture is a commonly employed *in vitro* system for investigating various neurodevelopmental stages, including axonal growth, dendritic arborization, and synapse formation and maintenance. We routinely prepare cultured neurons using mouse hippocampi at embryonic day ~18-19. These neurons first extend their axons and dendrites and then form mature dendritic spines at ~18 days *in vitro*. The cultures can be maintained for at least 4 weeks *in vitro*, though we usually try to complete all experiments within three weeks. The culture system allows us to investigate the involvement of genes in different neurodevelopmental stages, depending on how long the neurons have been cultured *in vitro*. Certainly, results derived from *in vitro* cultures should be validated *in vivo* using mouse genetic models. In our series of studies on CTTNBP2, we first demonstrated a role for CTTNBP2 in dendritic spine formation and maintenance using cultured hippocampal neurons (Chen and Hsueh, 2012, Journal of Neuroscience 32:1043; Shih et al., 2020, Acta

Neuropathologica Communications 8:185). Next, we established *Cttnbp2* knockout mice and knockin mice expressing ASD-linked CTTNBP2 mutant variants, including M120I, R533* and D570Y. We have observed synaptic defects (such as reduced number and size of dendritic spines) in all of the *Cttnbp2* knockout and mutant mice at ages appropriate for behavioral assays. We would like to emphasize that although synapse formation (including both pre- and post-synaptic terminals) represents a critical process of neuronal development, synapses are not fixed subcellular structures. Instead, they are dynamically regulated and undergo changes after they form and thereafter throughout their entire lifespan. Morphological shrinkage vs. enlargement, maintenance vs. elimination, and activity plasticity continuously impact synapses after they have formed. Thus, how synaptic ASD proteins control synaptic formation and function is not limited to the early developmental stage. Importantly, we previously showed that CTTNBP2 is also critical for dendritic spine maintenance (Chen and Hsueh, 2012, Journal of Neuroscience 32:1043), supporting a long-lasting impact of CTTNBP2 on synaptic number and size. Use of cultured neurons is widely accepted in the literature as an appropriate approach for investigating the impact of ASD-linked genes on synapse formation and maintenance, with mouse models then being employed to validate the results *in vivo*. Certainly, a system to monitor neuronal activity/morphology and link it to behavioral change throughout development would prove very useful for live-recording the features of ASD mice. It is a long-term goal that we should all aim for.

To understand which brain region(s) are particularly sensitive to *Cttnbp2* deficiency, we previously conducted C-FOS staining to monitor neuronal activity upon reciprocal social interaction. Our results show that the neuronal activity of the hippocampal CA1 and CA3 regions and basolateral amygdala of *Cttnbp2* knockout mice is reduced upon social stimulation compared with wild-type littermates (Shih et al., 2020, Cell Reports 31:107700). Since the hippocampus is a region known to be affected by *Cttnbp2* deficiency, these published results support the relevance of our analysis here using cultured hippocampal neurons. To date, it remains unclear which neural networks in the brain are affected by *Cttnbp2* mutations and this remains an important topic for further study, though we have already identified the hippocampus and amygdala as susceptible to *Cttnbp2* deficiency.

Our previous studies have indicated that CTTNBP2 acts as a multidomain adaptor molecule to control F-actin and microtubule cytoskeletons and synaptic signaling molecules. In immature neurons, CTTNBP2 is distributed along dendrites and it regulates dendritic arborization by bundling microtubules (Shih et al., 2014, Journal of Cell Science 127:3521). In mature neurons, CTTNBP2 interacts with cortactin to control dendritic spine formation and maintenance (Chen and Hsueh, 2012, Journal of Neuroscience 32:1043). Moreover, CTTNBP2 can either form homo-oligomers with itself or hetero-oligomers with striatins, a regulatory subunit in the protein phosphatase 2A family (Chen et al., 2012, Molecular Biology of Cell 23:4383), which may contribute to synaptic signaling. Recently, proteomic analysis further showed that *Cttnbp2* knockout alters the synaptic distribution of more than 100 proteins (Shih et al., 2020, Cell Reports 31:107700). Some CTTNBP2-regulated proteins

physically associate with CTTNBP2. However, it remains unclear how CTTNBP2 interacts with and controls the synaptic distribution of those proteins. In our current study, we show that CTTNBP2 aggregates in neurons are caused by LLPS. CTTNBP2 condensation represents a possible mechanism by which CTTNBP2 can effectively interact with and facilitate synaptic targeting of its interacting proteins.

We previously analyzed the behavioral features of *Cttnbp2* knockout mice and mutant mice carrying ASD-linked *Cttnbp2* mutations. We demonstrated that all *Cttnbp2*^{+/-}, *Cttnbp2*^{-/-}, *Cttnbp2*^{+/R533*} and *Cttnbp2*^{+/M120I} mice exhibited synaptopathy and social deficits (Shih et al., 2020, Cell Reports 31:107700; Shih et al., 2020, Acta Neuropathologica Communications 8:185), confirming a link between CTTNBP2-related synaptopathy and ASD. In the current study, we further show that ASD-linked M120I, R533* and D570Y mutations of *Cttnbp2* differentially influence CTTNBP2 condensate formation. These results provide evidence for the relevance of CTTNBP2 LLPS to dendritic spine formation and maintenance, as well as to ASD etiology.

Reviewer #1's specific comment #1: *In the introduction, a topic of SARS-CoV-2 is irrelevant.*

Our response: Our description about SARS-CoV-2 has now been removed from the Introduction of the revised manuscript.

Reviewer #1's specific comment #2: *Page 4, line 16 from bottom, a citation is needed.*

Our response: Reference #39 [Shih, P.Y. et al. 2020. CTTNBP2 controls synaptic expression of zinc-related autism-associated proteins and regulates synapse formation and autism-like behaviors. *Cell Reports* **31**, 107700] is now cited in the revised manuscript.

Reviewer #1's specific comment #3: *Page 5, line 2, a definition or explanation of IDP should be stated.*

Our response: Apologies, we now clarify that IDP stands for “intrinsically disordered protein” in the revised manuscript.

Reviewer #1's specific comment #4: *Page 6, line 13, It is mentioned that the results from Figure 1b and 1c indicate the formation of membraneless condensates. However, to prove it, evidence supporting the absence of membrane should be presented.*

Our response: We now include a set of FRAP experiments in **new Supplementary Fig. S2d**, in which we photobleached almost all GFP-CTTNBP2 signal of a single condensate in a COS1 cell. The GFP signal in the bleached condensate was rapidly increased after photobleaching and distributed across the entire condensate within a couple of minutes. In this set of experiments, the photobleached condensate was isolated from other condensates and we

did not observe fusion of that condensate with others during the recording period. Thus, recovery of CTTNBP2 signal likely arises from molecule exchange between the CTTNBP2 condensate and the neighboring cytoplasm. If the CTTNBP2 condensate was a membranous structure, we would expect the membrane to limit molecule exchange between the condensate and neighboring area, thereby abolishing fluorescence recovery after photobleaching. However, instead, we observed rapidly recovered fluorescence of CTTNBP2 condensates after photobleaching. Thus, although we have not directly investigated if CTTNBP2 condensates are membranous, this new data support that CTTNBP2 condensates are very unlikely to be membranous structures.

Reviewer #1's specific comment #5: Pages 5-6. Two different cell lines (i.e., Cos-1 and HEK293T) used here need to add explanation about general characteristics of CTTNBP2 proteins on these cell lines. It would be required to add another set of photos of condensate formation in HEK293T to Figure 1b/1c.

Our response: We now include a new set of results illustrating CTTNBP2 condensates in HEK293T cells. In new **Supplementary Fig. S2a**, we compared HEK293T cells transfected with GFP alone or GFP-CTTNBP2. Similar to our observations for COS1 cells subjected to the same treatments, CTTNBP2 (but not GFP) forms condensates in HEK293T cells. The CTTNBP2 condensates also rapidly moved and underwent repetitive fusion and fission in HEK293T cells. We also now include new **Supplementary Video S2**, in which we present a live recording that shows the movement of GFP-CTTNBP2 condensates in HEK293T cells, as well as new **Supplementary Fig. S2c** that shows the starting point (as a static image) of new Supplementary Video S2.

Reviewer #1's specific comment #6: Page 6, lines 25-26. Further explanation is required why the formation of spherical droplets in the presence of PEG support the "intrinsic" property of CTTNBP2 in relation to the condensate formation.

Our response: PEG, a commonly used crowding agent, is routinely added to solutions to mimic the crowded environment inside living cells (Phillip et al., 2009, Biophysical Journal 97:875). Given that PEG does not alter protein properties but mimics the condition of higher protein concentrations (Annunziata et al., 2002, PNAS 99:14165; Abbott et al., 1991, Macromolecules 24:4334; Phillip et al., 2009, Biophysical Journal 97:875), it has been used as an essential component in solution to demonstrate the LLPS of many proteins (McDonald et al., 2020, Nature 588:454; Milovanovic et al., 2018, Science 361:604; Pechstein et al., 2020, Cell Reports 30:2594; Singh et al., 2020, JBC 295:5850; Wegmann et al., 2018, EMBO J. 37:e98049). Accordingly, we adopted the commonly used approach of including PEG in our assay. To further confirm that condensate formation depends on CTTNBP2, we have included GFP as a negative control, which cannot form condensates even in the presence of PEG (**Fig. 1d**). In addition, GFP-CTTNBP2₁₋₂₇₁ that does not contain the C-terminal intrinsically

disordered region (IDR) lacks the ability to form condensates (**Fig. 1d**). Thus, the condensate formation ability of CTTNBP2 is determined by its C-terminal IDR. These results support that CTTNBP2 forms condensates. Nevertheless, we agree to remove “intrinsic” from page 6 line 26 to avoid unnecessary misunderstanding.

Reviewer #1's specific comment #7: Page 20, Fig 2b. Supernatant (S) and Pellet (P) fraction should be labeled in the figure.

Our response: We apologize for this oversight. The “S” and “P” fractions are now labeled in revised **Fig. 2b**. In addition, the original results of all uncropped gels and quantifications of band intensity have been included in **Supplementary Table S2 Raw data**.

Reviewer #1's specific comment #8: Figure 8f. Behavioral data. The data looks messed up here and difficult to understand what occurs here. Need to remake the figure and analysis.

Our response: We apologize for forgetting to include genotype information in original **Fig. 8f** (new **Fig. 9g** in the revised manuscript). In our revised version, we now specify that the blue dots represent the results of individual $Cttnbp2^{+/M120I}$ mice and the black dots are the data for WT littermates. The same mice underwent three consecutive trials of reciprocal social interaction (RSI) tests with one-week intervals between trials. The results for the same animals in all three trials are linked by blue or black lines, allowing us to monitor the behavioral changes of individual mice among different treatments. The first RSI test was performed when mice were ~9-week-old. Before the first RSI test, mice drank regular water (“H₂O” in the figure). The results indicate that $Cttnbp2^{+/M120I}$ mice displayed reduced social interaction compared with wild-type littermates. Directly after the first RSI test, regular water was substituted with water supplemented with 40 ppm zinc (“Zn²⁺” in the figure) for 1 week. The second RSI test was then conducted when mice were 10-week-old. Comparing the results of the first and second RSI tests, we observed that zinc supplementation enhanced the social interaction of $Cttnbp2^{+/M120I}$ mice but not their wild-type littermates. Without zinc supplementation, social interaction between $Cttnbp2^{+/M120I}$ mice and wild-type littermates was comparable. After the second RSI test, zinc-supplemented water was replaced by regular water before conducting the third RSI test one week later when the mice were 11-week-old. The third RSI test was designed to examine if the beneficial effect of zinc supplementation could be long-lasting. We found that four of the $Cttnbp2^{+/M120I}$ mice exhibited reduced social interaction upon zinc supplementation being withdrawn, but six of them still maintained their elevated social interaction. Thus, sensitivity to withdrawal of zinc supplementation varies among individuals. Nevertheless, this set of behavioral assays allows us to understand if zinc exerts a beneficial effect on mouse social behavior. We wish to emphasize that the same set of assays was used previously to analyze $Cttnbp2^{-/-}$ and $Cttnbp2^{+/R533*}$ mice. In that study, zinc supplementation enhanced zinc levels in the brains of those mouse lines (Shih et al., 2020,

Cell Reports 31:107700). Similar to *Ctnnbp2*^{+/^{M1201}} mice, that study demonstrated that zinc supplementation had a beneficial effect on the social behaviors of both *Ctnnbp2*^{-/-} and *Ctnnbp2*^{+/^{R533*}} mice (Shih et al., 2020, Cell Reports 31:107700). We now provide a more detailed description of these experiments on **page 15** of the revised manuscript.

We thank Reviewer #1 for her/his constructive suggestions and comments, which have given us the opportunity to add missing information and to clarify our points.

Reviewer #2's general comment #1: *In this study, authors examine the role of Cortactin-binding protein 2 (CTTNBP2) in dendritic spines. They find that purified CTTNBP2 undergoes PEG-dependent liquid-liquid phase separation (LLPS) via its intrinsically disordered C-terminal domain. CTTNBP2 also binds zinc using its N-terminal coiled-coil domain. Zn binding induces CTTNBP2 aggregation in vitro and in cells. They investigate the effect of Zn and 1,6-hexanediol on synaptic activation and synaptic retention of CTTNBP2 and SHANK3. They also examine the role of several Autism spectrum disorder mutations on CTTNBP2 phase separation and Zn dependent aggregation. The authors' data that CTTNBP2 binds Zn to regulate its higher order oligomerization and regulate its function is strong and compelling, but the evidence supporting a role for CTTNBP2 LLPS in neurons is lacking.*

Although the authors show that purified CTTNBP2 can undergo LLPS in vitro, the cellular experiments are not rigorous enough to conclude CTTNBP2 LLPS is important to the formation or function of CTTNBP2 aggregates in neurons. The main experiments to test for CTTNBP2 LLPS in neurons rely on 1,6-hexanediol, which could potentially cause many off-target effects or just broadly disrupt the PSD. The authors treat cells with 4-8% hexanediol for 25 min. With these conditions, many cellular condensates (including the entire postsynaptic density) could be dissolved. Furthermore, 1,6-hexanediol has been shown to have effects independent of LLPS, such as broadly inhibiting kinase and phosphatase activity (<https://doi.org/10.1016/j.jbc.2021.100260>). Finally, if treatment with 1,6-hexanediol was simply preventing CTTNBP2 LLPS, then the effect of hexanediol should be fully reversible, and CTTNBP2 condensates would reform rapidly after hexanediol washout. In Fig. 5e the authors show that hexanediol treatment in neurons is not reversible, which is not consistent with effects due to LLPS of CTTNBP2. With all of these concerns, it is troubling that the authors rely exclusively on sensitivity to 1,6-hexanediol as evidence for CTTNBP2 LLPS in neurons. Additional experiments, such as expressing CTTNBP2 mutations or truncations that specifically do not undergo LLPS would provide stronger evidence that CTTNBP2 LLPS is important in neurons.

Our response: We very much appreciate Reviewer #2's insightful comments and suggestions. In the revised manuscript, we now address this reviewer's concern regarding the effect of 1,6-HD and phase separation on CTTNBP2 in neurons through the following four arguments.

(1) Specificity of 1,6-HD treatment. We wish to clarify that we used 8% 1,6-HD in our crude lysate experiments, which effectively dissolved CTTNBP2 condensates in solution. However, we applied 4% 1,6-HD to cultured hippocampal neurons in our original manuscript because we did notice that the cultured neurons looked unhealthy when higher

amounts of 1,6-hexanediol were used. As Reviewer #2 points out, using 5-10% 1,6-HD for *in vitro* kinase or phosphatase assays will reduce the activities of the examined phosphatases and kinases. Accordingly, to minimize potential non-specific effects, we reduced the concentration of 1,6-HD to 4%. First, we examined the dendritic spine density to confirm that the neuronal morphology of treated hippocampal neurons was not affected by 4% 1,6-HD (**Figure A**). In contrast, we observed that the size of CTTNBP2 condensates was much reduced, and condensate number was increased, upon 4% 1,6-HD treatment (new **Fig. 7e-7f** in the revised manuscript). Under the same condition, the number of F-actin puncta was reduced, though their size was not affected, by 4% 1,6-hexanediol (new **Fig. 7e-7f**). As F-actin responds differentially to 4% 1,6-HD, the effect of 4% 1,6-HD treatment on CTTNBP2 is specific to some degree.

- (2) The reversible effect of 1,6-HD on CTTNBP2 condensates in immature neurons.** We have now revisited our 1,6-hexanediol (1,6-HD) washout experiment in cultured neurons. To deliver 1,6-HD into cells, digitonin, a mild detergent, has to be included to open the plasma membrane. Washout removes both 1,6-HD and digitonin from the medium. Therefore, intracellular 1,6-HD may not be freely released into extracellular solution after removing digitonin. The 1,6-HD remaining in cells may slow CTTNBP2 LLPS recovery. Given that long-term addition of digitonin in culture medium may harm neurons, it is problematic to modify this part of our experiment. However, we did notice that in our protocol we directly transferred coverslips with cultured neurons back into growth medium after the 1,6-hexanediol treatment. Thus, the residual 1,6-hexanediol solution on the coverslip was transferred into growth medium and likely influenced the effect of washout. Accordingly, we have modified our procedure by removing the residual solution on the coverslips as completely as possible, then rinsing the coverslips once with medium, before placing the coverslip back into the culture medium for recovery. After recovery for 0, 20, 40 or 60 min, we fixed the neurons, performed immunostaining, and determined the aggregation index of CTTNBP2. Our new results show that the aggregation indices of CTTNBP2 were reduced after 1,6-HD treatment and gradually increased after 1,6-hexanediol washout. Comparisons between the “0” vs. “40” and “0” vs. “60” groups revealed significant differences. Moreover, aggregation levels at 40 and 60 min after 1,6-HD washout were comparable to the control without 1,6-HD treatment. These results show that the effect of 1,6-HD on CTTNBP2 condensates is reversible in immature neurons. We would like to emphasize that our experiments were routinely performed in a blind fashion by having another lab member relabel the samples. This approach minimizes the effect of personal bias. We thank Reviewer #2 for this criticism, which gave us the opportunity to

conduct a more rigorous experiment and to revise this critical conclusion. These new results have been included as **new Fig. 6** of the revised manuscript.

(3) The reversible effect of 1,6-HD on CTTNBP2 condensates in mature neurons. The above-described 1,6-HD washout experiment revealed that the effect of 1,6-HD on CTTNBP2 LLPS is reversible in immature neurons. Given that the synaptic distribution of CTTNBP2 and its regulated proteins are critical for brain function and social behaviors (Shih et al., 2020, Cell Reports 31:107700), we assert that investigating CTTNBP2 condensates in dendritic spines is more critical. Thus, we designed the second washout experiment in the revised manuscript to examine the effect of 1,6-HD on CTTNBP2 in mature neurons. Similar to our approach for immature neurons, we used 4% 1,6-HD to treat mature neurons at 18 days in vitro and then washed it out before recovery for 0, 20, 40 or 60 min. Immunostaining using CTTNBP2 antibody and phalloidin was performed to monitor the distribution and aggregation of endogenous CTTNBP2 and F-actin. F-actin is highly concentrated at dendritic spines, the subcellular structure of excitatory synapses. As observed for immature neurons, 1,6-HD treatment dissolved CTTNBP2 condensates and resulted in them being smaller, but enhanced their number. Thus, the 4% 1,6-HD treatment appeared to partially dissolve CTTNBP2 condensates. The response of CTTNBP2 condensates to 1,6-HD was somewhat specific because the effect of 1,6-HD on F-actin was different in that the size of F-actin puncta was not obviously altered but their number was reduced. Moreover, after 1,6-HD washout for 40 or 60 min, the size of CTTNBP2 condensates significantly increased relative to those without 1,6-HD treatment, though they remained smaller than the control without 1,6-HD treatment. In contrast, F-actin puncta remained small and their numbers did not recover after washout. Thus, the effect of 1,6-HD on CTTNBP2 LLPS is also reversible in mature neurons. However, given that the response of F-actin to 1,6-HD was not similar to that of CTTNBP2, the effect of 1,6-HD on CTTNBP2 appears to be somewhat specific. These new results of our 1,6-HD washout experiment on mature neurons have been included in the revised manuscript as **new Fig. 7e-7f**.

(4) Ctnb2 mutations influence LLPS. We completely agree with Reviewer #2 that investigating a mutation that specifically influences CTTNBP2 LLPS would be a nice addition to our study. Forming condensates is an effective mechanism for clustering functionally related proteins. The LLPS mediated by an intrinsically disordered region is achieved by accumulated weak hydrophobic interactions. Nevertheless, predicting a specific mutation that would disrupt such accumulative weak interactions is challenging. LLPS of CTTNBP2 is mediated by its C-terminal intrinsically disordered region, encompassing ~two-thirds of the CTTNBP2 protein. In addition to forming condensates, we have shown previously that the C-terminal intrinsically disordered region of CTTNBP2 is also involved in microtubule association and cortactin binding (Chen and Hsueh, 2012,

Journal of Neuroscience 32:1043; Shih et al., 2014, Journal of Cell Science 127:3521). Under this scenario, CTTNBP2 likely condenses and clusters with its interacting proteins via its C-terminal regions. Therefore, it is very difficult to predict a mutation that would specifically affect LLPS while not influencing other interactions. Thus, we examined the spontaneous mutations that have been identified in patients with ASD. We showed previously that three ASD-linked CTTNBP2 mutant variants, i.e. M120I, R533* and D570Y, alter how the protein regulates dendritic spine formation. Accordingly, if LLPS is a critical feature of CTTNBP2, could certain ASD-linked mutations influence CTTNBP2 LLPS? In the current study, we indeed found that all three of these ASD-linked mutations impact CTTNBP2 LLPS and influence the synaptic distribution of the protein. Among these three mutant variants, the **M120I** mutation does not directly influence the interaction of CTTNBP2 with other proteins, but does reduce the N-C terminal self-interaction of CTTNBP2, thereby impairing CTTNBP2 LLPS. Thus, the M120I mutant could be a representative variant that primarily impairs LLPS and therefore controls the synaptic distribution and retention of CTTNBP2. Our study suggests that CTTNBP2 condensation is sensitive to ASD-linked mutations. Moreover, it strengthens the significance of LLPS in the protein's function and evidences a link between LLPS and ASD. The results on these ASD-linked mutations have been included in new **Figures 3, 4, 5, and 9** of our revised manuscript.

Among the three ASD-linked mutations we examined, we show in the current manuscript that the **D570Y** mutant formed more stable condensates in COS1 cells and neurons and it was retained more at dendritic spines upon 1,6-HD treatment (new **Fig. 3b, 9**). CTTNBP2 D570Y mutant protein exhibited stronger protein-protein interactions and a propensity to form condensates, and was more resistant to 1,6-HD treatment (new **Figure 9b, 9c**). Although CTTNBP2 cannot be the only protein affected by 1,6-HD, the resistance of the D570Y mutant to 1,6-HD is also consistent with our observation that 1,6-HD acts on CTTNBP2 to regulate condensate formation and its synaptic distribution.

The **R533*** mutant that lacks the extreme C-terminal 98 amino acid residues of CTTNBP2 was previously shown to display impaired synaptic CTTNBP2 targeting in mature neurons (Shih et al., 2020, Acta Neuropathologica Communications 8:185). In the current study, we have further shown that the R533* mutation reduces CTTNBP2 condensation in solution (**Fig. 4, 5**), consistent with a role for C-terminal-mediated LLPS in synaptic CTTNBP2 targeting. To make this point even clearer, we repeated the immunostaining experiment for the revised manuscript to show that R533* mutant proteins are quite evenly distributed in the soma and dendritic shaft but not in dendritic spines. This new data has been included as **new Fig. 9a**.

However, we must add that the R533* mutant also lacks the critical proline residues P540 and P543 in the cortactin-binding site (Chen and Hsueh, 2012, Journal of Neuroscience 32:1043). Thus, based on our results, we cannot be absolutely sure that the defects caused by R533* mutation are solely due to impaired LLPS. In response to Reviewer #2's suggestion, we have now also considered a mutant variant, **S550***, that still binds cortactin by retaining the cortactin-binding site but lacks the C-terminal tail (81 amino acids). Indeed, we found that the S550* mutant still interacts with cortactin, albeit with an affinity seemingly lower than for WT (**Fig. Ba**). Similar to our observations for the

R533* mutant, the S550* mutant was widely distributed in the soma and dendrites. Although the S550* mutant was still present in dendritic spines, it was not highly enriched there, suggesting that cortactin binding is insufficient for dendritic spine enrichment (**Fig. Bb**), supporting that the last 81 amino acid residues of CTTNBP2 are critical for its synaptic targeting. However, unexpectedly, we observed that the S550* mutant was not evenly distributed in COS1 cells, displaying a microtubule-associated pattern in COS1 cells (**Fig. Bc**). Given that microtubules are the major cytoskeletons in dendritic shafts, we cannot be sure if reduced dendritic spine targeting of mutant CTTNBP2 is caused by lower LLPS activity or higher affinity for microtubules or both.

Apart from the R533* and S550* mutants, we did include a CTTNBP2₁₋₂₇₁ truncated fragment in our original manuscript. The CTTNBP2₁₋₂₇₁ truncated fragment was evenly distributed in the cytoplasm of COS1 cells (**Fig. 1b**). It did not form condensate in solution (**Fig. 1d**). Our previous study also showed that the N-terminal coiled-coil domain was evenly distributed in the dendritic shaft but not in dendritic spines (Shih et al., 2014, Journal of Cell Science 127:3521). Although these results on R533*, S550* and the CTTNBP2₁₋₂₇₁ fragment cannot completely rule out an involvement of microtubule and/or cortactin, they are at least consistent with our model that LLPS contributes to the dendritic spine targeting and retention of CTTNBP2.

The **M120I** mutation exhibited a very intriguing effect on condensate formation. The M120I mutation is located within the N-terminal coiled-coil domain of CTTNBP2. Our previous study showed that this mutation does not influence the examined protein-protein interactions of CTTNBP2 that are mediated via the N-terminal coiled-coil region. Instead, it impairs the interaction between the N-terminal coiled-coil region and the C-terminal proline-rich domain of CTTNBP2 (Shih et al., 2020, Acta Neuropathologica Communications 8:185). In our current study, we found that the M120I mutation reduces the size and number of droplets in solution (**new Fig. 4, 5**). In COS1 cells, the M120I mutant can still form condensates but is much more mobile (**Fig. 3c**). In neurons, the M120I mutation also resulted in higher sensitivity to 1,6-HD compared to WT CTTNBP2, which is reflected in its diminished distribution in dendritic spines after 1,6-HD treatment (**new Fig. 9d-9e**). Thus, although the M120I mutation is located in the N-terminal region of CTTNBP2, it influences LLPS and reduces the condensate formation of CTTNBP2 likely by impairing N-C-terminal self-interactions. Since the M120I mutation does not directly disrupt the known protein-protein interactions of CTTNBP2, it is the most specific mutation of CTTNBP2 that impairs CTTNBP2 condensate formation.

In conclusion, all five of the selected *Cttnbp2* mutants we assessed, including M120I, R533*, D570Y, S550* and CTTNBP2₁₋₂₇₁, differentially impact LLPS and the synaptic distribution of CTTNBP2. We have now extensively rewritten the Introduction, Results and Discussion sections to make our point clearer. We thank Reviewer #2 for her/his comments and suggestions.

Reviewer #2's general comment #2: Their data supports the model that Zinc promotes higher-order CTTNBP2 oligomerization, which drives the formation of solid aggregates. They claim that "LLPS is the mechanism by which CTTNBP2 condenses in dendritic spines of cultured hippocampal neurons", but their data does not definitively support this conclusion. Their data more strongly supports the model that Zn-dependent oligomerization and aggregation of CTTNBP2 occurs independently of CTTNBP2 LLPS. The authors show that Zn induced aggregate occurs in the absence of PEG (Fig. S2c). However, LLPS requires PEG (Fig. 1i). These observations suggest that Zn induced CTTNBP2 aggregation does not depend upon CTTNBP2 LLPS in vitro (i.e. Zn induced aggregation does not depend upon a prior LLPS, but can occur de novo as a distinct oligomerization pathway). Similarly, residues 104-271 are sufficient for Zn-dependent oligomerization in vitro (Fig. 2), but a similar fragment (1-271) does not undergo LLPS in vitro (Fig. 1). This also suggests that oligomerization occurs independently to LLPS, not following LLPS. Their data about Zn-dependent oligomerization and aggregation in vitro and in neurons is novel and would be interesting without the need to invoke LLPS.

Our response: We thank Reviewer #2 for these comments and her/his recognition of our assessment of the effect of zinc on CTTNBP2. Nevertheless, we feel Reviewer #2's comments reflect a misunderstanding of the link between zinc and LLPS. Accordingly, we have

extensively rephrased the relevant parts of our revised manuscript and added more results to clarify our points. Before addressing the reviewer's specific questions and comments, we feel it would be helpful if we first define the terminology we use in the following paragraphs and in the revised manuscript.

(1) "Oligomerization" means homophilic interaction via the N-terminal coiled-coil domain of CTTNBP2. It may result in CTTNBP2 dimer, trimer or tetramer. This property is determined by the amino acid sequence. (2) "High-order multimerization" means that these CTTNBP2 oligomers further interact with each other via their association with zinc, potentially generating huge aggregates or even precipitating CTTNBP2 out of solution. (3) "Liquid-liquid phase separation (LLPS)" means that CTTNBP2 remains in liquid phase even though it condenses to form droplets and separates from solution. (4) The "Liquid-to-gel phase transition" describes the phase change induced by binding of zinc to CTTNBP2. Zinc binding induces high-order multimerization or even precipitation of CTTNBP2 oligomers and greatly reduces the mobility of CTTNBP2 in condensates.

We completely agree with Reviewer #2 that zinc binding and LLPS can occur independently. Based on our study, we suggest that CTTNBP2 undergoes LLPS via its C-terminal intrinsically disordered region. Zinc binds to the N-terminal coiled-coil region to induce high-order multimerization of CTTNBP2 oligomers. When these two types of interactions occur at the same time, CTTNBP2 condensates can be further clustered (**new Fig. 2h-2i**) and the mobility of CTTNBP2 proteins in the condensates are also greatly reduced, resulting in a liquid-to-gel phase transition (**Fig. 3**). As the effect of zinc on the liquid-to-gel phase transition of CTTNBP2 is reversible (**Fig. 2i**, also see **our response to Reviewer #2's specific comment #8**), the protein phase of CTTNBP2 can be dynamically controlled by the concentration of zinc. Given that postsynaptic zinc mainly comes from presynaptic vesicles, levels of postsynaptic zinc are controlled by synaptic activation. This scenario suggests that the liquid-to-gel phase transition of CTTNBP2 may be regulated by neuronal activity. Gel-phase CTTNBP2 is likely more resistant to 1,6-HD treatment than liquid-phase CTTNBP2. To address these issues, we performed several different experiments for the original manuscript. First, we showed that addition of zinc results in more CTTNBP2 being retained in dendritic spines upon 1,6-HD treatment (**new Fig. 7a-7d**), consistent with our result that zinc reduces CTTNBP2 mobility in COS1 cells (**Fig. 3**). Second, we used bicuculline to enhance the neuronal activity of cultured neurons. Our results showed that bicuculline treatment also resulted in more CTTNBP2 being retained at dendritic spines upon 1,6-HD treatment (**new Fig. 7g-7j**). The effect of bicuculline is mediated by zinc because addition of TPEN attenuated the effect of bicuculline treatment (**new Fig. 7k-7m**). Together, these results support that CTTNBP2 undergoes LLPS to form condensates in the dendritic spines of mature neurons and that CTTNBP2 condensates can be further immobilized by zinc, an ion co-released with glutamate from presynaptic vesicles.

In addition, we have also re-investigated the effect of zinc on CTTNBP2 condensate formation in immature neurons. Zinc supplementation maintained CTTNBP2 condensates in the presence of 1,6-HD treatment in immature neurons, as aggregation indices of CTTNBP2 were not significantly altered by 1,6-HD when extra zinc was present. These new results have been included in the revised manuscript as **Fig. 6f-6g**. These new results also suggest that the high-order multimerization of CTTNBP2 oligomers mediated by zinc binding may nucleate CTTNBP2 oligomers and stabilize CTTNBP2 condensates.

Thus, our study reveals that zinc binding and LLPS work together to control the protein phase, synaptic distribution and protein-protein interactions of CTTNBP2 (such as its interaction with SHANK3) in neurons. By incorporating both zinc binding and LLPS experiments in our current work, we comprehensively reflect the complex regulation overseeing CTTNBP2 activity in neurons. We have now extensively rephrased the respective part of our manuscript. We hope that Reviewer #2 is satisfied by our explanations and the new results we present in the revised manuscript.

***Reviewer #2's specific comment #1:** They often show a zoomed up image of a single condensate or small group of condensates from cellular experiments (e.g. Fig 1c, Fig 3). In these figures, could the authors also include an image of the entire cell and highlight the ROI containing the condensates in the zoomed images? It is hard to understand these images without the context of the cell. For example, 1c looks very different than 1b, so it would be helpful to see how these condensates appear in the whole cell.*

Our response: In live-recording experiments using an LSM980 system with Airyscan 2, we do not usually record images of the “entire” cells because it takes much longer to record and process the images. Instead, we select a region covering ~one-third of the cells. In the revised manuscript, we have now included images encompassing a larger area (**new Supplementary Fig. S5a**). We also wish to emphasize that **Fig. 1b** and **1c** are not the results of the same cells. In **Fig. 1b**, the samples were fixed COS1 cells, aimed at generating high-resolution images. In **Fig. 1c**, the settings favor high temporal resolution for live-recording.

***Reviewer #2's specific comment #2:** In Fig. 2b S and P are not labeled in the figure.*

Our response: We apologize for the oversight and have now labeled the “S” and “P” fractions in revised **Fig. 2b**.

***Reviewer #2's specific comment #3:** Do authors limit or measure the amount of overexpression? The two phenotypes they see in Fig. S3b could be explained by different expression levels. Often one can tell from integrated GFP intensities if cells are expressing lower or higher levels of protein. Since LLPS is highly concentration dependent, different cytoplasmic levels of overexpressed proteins could lead to different phenotypes.*

Our response: Our previous studies have indicated that CTTNBP2 binds and bundles microtubules in cells (Shih et al., 2014, Journal of Cell Science 127:3521). The D570Y mutation increases the association of CTTNBP2 with microtubules (Shih et al., 2020, Acta Neuropathologica Communications 8:185). In **new Fig. S4b**, we have tried to illustrate that the D570Y mutant still forms condensates, despite the enhanced microtubule association of D570Y mutant-expressing cells. WT CTTNBP2 also exhibited a microtubule-associated distribution pattern in some cells. As Reviewer #2 has suggested, we have now quantified the intensity of WT and D570Y mutant signals. When we pooled all cells, we found that signal intensities of WT and D570Y were comparable. When we separated cells based on microtubule association, we found that cells displaying a microtubule-associated CTTNBP2 distribution pattern tended to have a higher level of CTTNBP2, regardless of whether we were assessing WT or D570Y mutant protein. Expression levels of CTTNBP2 are likely relevant to its microtubule association. Certainly, the condensate formation of CTTNBP2 in COS1 cells is also relevant to its expression levels, as it depends on protein concentration in solution (**Fig. 1g-1i**). However, since COS1 cells represent an ectopic expression system, we think it is more meaningful to investigate CTTNBP2 in neurons and, accordingly, have not placed too much emphasis on our COS1 experiments.

Reviewer #2's specific comment #4: In Fig. 4, the authors claim that ASD mutations impact phase separation of CTTNBP2. However, these experiments are done in crude cell lysates, thus one cannot rule out the possibility that these mutations are changing interactions with unknown proteins in the lysate. Experiments with purified protein would be required to conclude that these mutations directly change the LLPS propensity of the protein.

Our response: For the revised manuscript, we have expressed and purified GFP-tagged M120I, R533* and D570Y mutant proteins and used these purified GFP-tagged proteins in condensate formation assays. We found that the purified GFP-M120I, -R533* and -D570Y mutants all formed smaller and fewer condensates in solution (**new Fig. 4e-4i** in the revised manuscript). For the M120I and R533* mutants, the results from purified proteins are consistent with those from crude lysates. In contrast, we obtained the opposite for the D570Y mutant. Since the D570Y mutant exhibits a higher affinity for binding microtubules (Shih et al., 2020, Acta Neuropathologica Communications 8:185), diverse cellular proteins in the crude lysates may be recruited to the D570Y condensates and thereby increase their size. Nevertheless, these new data support that the ASD-linked mutations of CTTNBP2 directly affect CTTNBP2 phase separation. Together with our results from cultured neurons (**new Figs. 7, 9**), we suggest that CTTNBP2 forms condensates at dendritic spines and that ASD-linked CTTNBP2 mutations alter its phase separation and synaptic targeting.

Reviewer #2's specific comment #5: The conclude that Shank3 and co-condense. Co-condensing would mean the proteins undergo phase separation together, but the authors are only quantifying co-localization.

Our response: Indeed, we have demonstrated co-localization of CTTNBP2 and SHANK3 in cultured neurons in original **Fig. 7 (new Fig. 8)**. However, Reviewer #2 may have overlooked original **Fig. 7a (new Fig. 8a)**, in which we demonstrated co-condensation of SHANK3 and CTTNBP2 by mixing purified GFP-CTTNBP2 (4 μ M) with crude lysate expressing SHANK3. SHANK3 alone forms small aggregates in solution (**new Fig. 8a**, upper, green particles). In the presence of high CTTNBP2 protein amounts (such as 4 μ M), CTTNBP2 formed huge condensates (red in **new Fig. 8a**) and recruited SHANK3 into the aggregates, with signal of SHANK3 aggregates outside of the CTTNBP2 condensates being much reduced (**new Fig. 8a**, middle and lower). Thus, SHANK3 co-condenses with CTTNBP2. To further confirm this point, we have now included the following two sets of new results in the revised manuscript:

(1) **CTTNBP2 and SHANK3 are present in the same condensates in COS1 cells.** We co-transfected CTTNBP2 and SHANK3 into COS1 cells and performed immunostaining to monitor the distribution pattern of both CTTNBP2 and SHANK3. All of the CTTNBP2 condensates also contained SHANK3 (**new Supplementary Fig. S6**), supporting that CTTNBP2 and SHANK3 co-condense under cellular conditions.

(2) **CTTNBP2 and SHANK3 co-condense in crude lysates.** We further transfected CTTNBP2 and SHANK3 into HEK293T cells and used the crude lysates to characterize condensate formation. When CTTNBP2 and SHANK3 were singularly transfected into HEK293T cells, they individually formed condensates in crude lysates. When CTTNBP2 and SHANK3 were co-transfected, they were present in the same condensates in crude lysates. Based on the colocalization coefficient, $\sim 42\% \pm 3.8\%$ of CTTNBP2 formed condensates with SHANK3. Compared with CTTNBP2 alone, mixing CTTNBP2 with SHANK3 increased the size and number of CTTNBP2 condensates. For SHANK3, co-existing CTTNBP2 also increased the size and number of SHANK3 condensates. These new results (**new Fig. 8b-8c** in the revised manuscript) further strengthen the evidence for co-condensation of CTTNBP2 and SHANK3 in solution. Our previous study also demonstrated co-immunoprecipitation of CTTNBP2 and SHANK3 from both mouse brains and transfected cells (Shih et al., 2020, Cell Reports 31:107700). Together, these findings support that CTTNBP2 and SHANK3 interact with each other and co-condense in dendritic spines in response to synaptic stimulation.

Reviewer #2's specific comment #6: Fig. 8f is missing critical information in the figure legend, so it is not possible to understand the data. The black lines and blue lines are two different conditions, but I can't tell what they are from the figure information. Please label the figure appropriately.

Our response: We apologize for forgetting to include the genotype information in original **Fig. 8f**. In the revised version, we have indicated that blue dots are the results of individual *Cttnbp2*^{+/^{M120I}} mice and that black dots are the data of WT littermates. The same mice

underwent three consecutive trials of reciprocal social interaction (RSI) tests with a one-week interval between trials. The results of the same animals in the three trials are linked by blue or black lines. These details and genotypes of examined mice have now been included on **Page 15** and new Figure 9g. We appreciate Reviewer #2 for highlighting this oversight.

Reviewer #2's specific comment #7: Does supplementing Zn in the diet actually change the amount of Zn in neurons or brain tissue? The amount of Zinc in the diet only doubles (84 ppm to 150 ppm) and it's not clear that that would necessarily increase the amount of Zn in neurons. Have the authors or other studies measured the effect of Zn dietary supplements on brain Zn levels?

Our response: Yes, dietary zinc does alter zinc levels in the brain. We have previously demonstrated that zinc supplementation for 7 days increases zinc levels in the brains of mice. Doubling zinc intake is sufficient to ameliorate synaptic targeting of CTTNBP2-regulated proteins and improve social behaviors of *Cttnbp2*^{-/-} and *Cttnbp2*^{+/^{R533}*} mice (Shih et al., 2020, [Cell Reports 31:107700](https://doi.org/10.1016/j.cellrep.2020.107700)). To make this point clearer, we now mention this previous study in the revised manuscript (**page 15**).

Reviewer #2's specific comment #8: What happens to Zn-induced aggregates once Zn is removed? This seems to be an important question for their model in figure 8. Zn-induced CTTNBP2 aggregates are solid like and quite stable, so it's possible their turnover would be very slow. Is Zn required to maintain oligomers over time, or is Zn only required for their initial nucleation? If Zn is removed do the aggregates immediately go away, or are they stable over longer periods of time? The authors could assess this in COS cells by treating cells with Zn and then washing out the Zn. If aggregates remained stable after Zn is removed, this could be an interesting mechanism for molecular memory of high Zn.

Our response: To explore if the effect of zinc on CTTNBP2 aggregation is reversible, we designed a zinc withdrawal experiment using TPEN, a zinc chelator, to treat zinc-GFP-CTTNBP2 aggregates. The experiment focused on three groups: (1) purified GFP-CTTNBP2 proteins (400 nM) plus PEG (3%); (2) zinc (10 μ M) added to a mixture containing equal amounts of purified GFP-CTTNBP2 proteins and PEG; (3) 30 mins after adding zinc to purified GFP-CTTNBP2 plus PEG, TPEN (15 μ M) was added into the mixture and incubated for a further 30 min. As the $K_{d_{Zn}}$ of TPEN is at the fM level (Huang et al., 2013, *Metallomics*. 5: 648–655) and the $K_{d_{Zn}}$ of CTTNBP2 is at μ M level, TPEN is expected to effectively compete for zinc binding with CTTNBP2 and remove zinc from CTTNBP2 multimers. Indeed, we found that addition of TPEN impaired the propensity of CTTNBP2 to form irregular condensates, instead forming droplet-like condensates. This outcome suggests that the effect of zinc on the liquid-to-gel phase transition of CTTNBP2 is reversible. These new data have been included in the revised manuscript as **new Fig. 2i**. Although the effect of zinc is reversible, it does not mean that zinc binding by CTTNBP2 cannot play a role in molecular memory. The gel phase or liquid-to-gel phase transition of CTTNBP2 may initiate a chain

reaction to further alter the molecular features of synapses and contribute to memory. Certainly, further investigations are required to explore that possibility. We anticipate studying this topic further in the future.

REVIEWERS' COMMENTS

Reviewer #1 (Remarks to the Author):

The manuscript has been revised nicely in accordance with responding well to the reviewers' request. I am satisfied with the inclusion of the original manuscript, which I thought reader the paper more valuable in terms of disease-related mechanisms.

Only minor changes I would request to include are

Line 552, page 18, "very useful in future for live-recording the features of ASD mice" should be changed like to "the features of ASD-relevant gene mutant mice".

All mutant mice are created with a target-mutation relevant to ASD, which is a candidate gene for ASD, but we do not have any mice possess ASD per se.

Line 564, page 18, "how zinc deficiency may sensitize patients to ASD-linked genetic variation" should be changed to "how zinc deficiency may aggravate ASD-linked genetic". There is similar reason I mentioned above. We did not assess ASD patients per se.

Reviewer #2 (Remarks to the Author):

The authors revised the manuscript to include additional experiments that address many of my initial concerns. They more rigorously demonstrate CTTNBP2 is sufficient to undergo phase separation using purified proteins in addition to the crude lysate experiments and demonstrate co-condensation of purified Shank and CTTNBP2. They also were able to show reversibility of condensate formation in cells upon treatment with hexanediol. The authors have clarified some of their language in the paper, making a clearer distinction between multimerization, liquid-liquid phase separation, and liquid-gel transition. They have corrected their mouse social experiment figure, such that readers can now understand the results. Their data shows that ASD mutations alter the LLPS behavior of CTTNBP2. The regulation of the liquid-gel transition by zinc is a novel result with intriguing implications. The study relies on correlations between the effects of point mutations and perturbations in vitro to the effects observed in cells. The final sentence of their abstract is a bit overstated "Our study evidences the relevance of condensate formation and zinc-induced phase transition to the synaptic distribution and function of ASD-linked proteins." I would prefer language like "Our study suggests the relevance of..."

There is section titled "LLPS and zinc influence synaptic CTTNBP2-SHANK3 interaction". Their data looks at CTTNBP2-SHANK3 association and colocalization, but doesn't directly explore the direct interaction between these two proteins. The original title of this section said "CTTNBP2-SHANK3 association" which is a more accurate description of their data.

The title also uses the word interaction. To me, interaction has a very specific meaning of two molecules directly binding each other with a measurable binding affinity. However, their study doesn't ever examine the direct interaction between CTTNBP2 and SHANK3, but rather looks at their localization relative to each other in vitro and in neurons. A better word to use in the title would be association or colocalization.

Our point-by-point response to Reviewers

Reviewer #1

The manuscript has been revised nicely in accordance with responding well to the reviewers' request. I am satisfied with the inclusion of the original manuscript, which I thought reader the paper more valuable in terms of disease-related mechanisms.

Only minor changes I would request to include are Line 552, page 18, "very useful in future for live-recording the features of ASD mice" should be changed like to the features of ASD-relevant gene mutant mice". All mutant mice are created with a target-mutation relevant to ASD, which is a candidate gene for ASD, but we do not have any mice possess ASD per se. Line 564, page 18, "how zinc deficiency may sensitize patients to ASD-linked genetic variation" should be changed to "how zinc deficiency may aggravate ASD-linked genetic". There is similar reason I mentioned above. We did not assess ASD patients per se.

Our response: We very much appreciate Reviewer #1's constructive suggestions for both rounds of this manuscript review process. Based on these new suggestions, we have changed the sentences on page 18 to: "Certainly, a system, such as a recently published mouse model⁶⁵, to monitor neuronal activity/morphology and link it to behavioral change throughout development would prove very useful in future for live-recording the features of ASD-relevant gene mutant mice." and "Our study represents an example of how LLPS and zinc supplementation integrate to control an ASD-linked postsynaptic protein and reveal a mechanism underlying how zinc deficiency may aggravate ASD-linked genetic variation and synaptopathy."

Reviewer #2

The authors revised the manuscript to include additional experiments that address many of my initial concerns. They more rigorously demonstrate CTTNBP2 is sufficient to undergo phase separation using purified proteins in addition to the crude lysate experiments and demonstrate co-condensation of purified Shank and CTTNBP2. They also were able to show reversibility of condensate formation in cells upon treatment with hexanediol. The authors have clarified some of their language in the paper, making a clearer distinction between multimerization, liquid-liquid phase separation, and liquid-gel transition. They have corrected their mouse social experiment figure, such that readers can now understand the results. Their data shows that ASD mutations alter the LLPS behavior of CTTNBP2. The regulation of the liquid-gel transition by zinc is a novel result with intriguing implications. The study relies on correlations between the effects of point mutations and perturbations in vitro to the effects observed in cells. The final sentence of their abstract is a bit overstated "Our study evidences the relevance of condensate formation and zinc-induced phase transition to the synaptic distribution and function of ASD-linked proteins." I would prefer language like "Our study suggests the relevance of..."

There is section titled "LLPS and zinc influence synaptic CTTNBP2-SHANK3 interaction". Their data looks at CTTNBP2-SHANK3 association and colocalization, but doesn't directly explore the direct interaction between these two proteins. The original title of this section said "CTTNBP2-SHANK3 association" which is a more accurate description of their data.

The title also uses the word interaction. To me, interaction has a very specific meaning of two molecules directly binding each other with a measurable binding affinity. However, their study doesn't ever examine the direct interaction between CTTNBP2 and SHANK3, but rather looks at their localization relative to each other in vitro and in neurons. A better word to use in the title would be association or colocalization.

Our response: We are very grateful for Reviewer #2's thoughtful and constructive suggestions. Accordingly, we have changed the manuscript title for this second revision to: "Phase separation and zinc-induced transition modulate synaptic distribution and association of

autism-linked CTTNBP2 and SHANK3”. The last sentence of our Abstract is now: “Our study suggests the relevance of condensate formation and zinc-induced phase transition to the synaptic distribution and function of ASD-linked proteins.” The title of the subsection on page 13 has been replaced with “LLPS and zinc influence synaptic CTTNBP2-SHANK3 association”.